

# Zooplankton diel vertical migration in the Corsica Channel (north-western Mediterranean Sea) detected by a moored ADCP

Davide Guerra[1], Katrin Schroeder[1], Mireno Borghini[1], Elisa Camatti[1], Marco Pansera[1], Anna Schroeder[1,2], Stefania Sparnocchia[1], Jacopo Chiggiato[1]

[1]Consiglio Nazionale delle Ricerche, Istituto di Scienze Marine (CNR-ISMAR), Venice 30122, La Spezia 19036, Trieste 34149, Italy
[2]University of Trieste, Faculty of Environmental Life Science, Trieste 34127, Italy

*Correspondence to*: Katrin Schroeder (katrin.schroeder@ismar.cnr.it)

**Abstract.** Diel vertical migration (DVM) is a survival strategy adopted by zooplankton, that was investigated in the Corsica Channel using ADCP data, from April 2014 to November 2016. The principal aim of the study is to characterize migratory patterns and biomass temporal evolution along the water column. The ADCP measured vertical velocity and echo intensity in the water column range between about 70 m and 390 m (the bottom depth is 443 m). In addition, net samples were taken during summer 2015 at the same location. During the investigated period, biomass had a well-defined daily and seasonal cycle, with peaks occurring in late winter – spring, when the stratification of water column is weaker. Biomass evolution along the whole water column is well correlated with primary production estimated with satellite data. Blooming and no-blooming periods have been identified and studied separately. During the no-blooming period biomass was most abundant in the surface and the deep layers, while during the blooming period the surface maximum disappeared and the deep layer with high biomass became thicker. These two layers are likely to correspond to two different zooplanktonic communities. Nocturnal DVM appears to be the main pattern during both periods, but also reverse and twilight migration are detected. Nocturnal DVM was more evident at mid-water than near in the deep and the surface layers. DVM occurred with different intensities in blooming and no-blooming periods, and phenomena like nocturnal sinking were found to be stronger during the blooming period. One of the main outcomes is that the principal drivers for DVM are light intensity and stratification, but also others are taken in consideration.

## 1 Introduction

Diel vertical migration (DVM) is one of the most important survival strategies adopted by zooplankton. During migration these marine organisms can cover vertical distances of a few hundred meters. At dawn zooplankton descends and remains at depth, where the probability of being predated by a visually hunting predator is lower; at dusk zooplankton rises to the euphotic layer and stays there during night to feed on phytoplankton (Ringelberg, 2009; Zaret and Suffern, 1976). This so called nocturnal migration is only one of the three most common migration patterns. Indeed, also twilight migration (ascent at dusk and sunrise, descent at midnight and immediately after sunrise) and reverse migration (ascent at sunrise, descent at sunset) have been described in previous studies (Haney, 1988 and references therein). The descent that occurs during night is called





midnight or nocturnal sinking and is a downward movement accomplished after the sunset ascent and before the sunrise descent, which some zooplanktonic organisms do to leave the surface feeding layer and return to depth (Pearre, 2003 and references therein). Indeed, many authors agree on the presence of a continuum of migrating behaviours between the two opposed patterns of nocturnal and reversed migration (Haney, 1988). Essentially, in nocturnal DVM, the benefit of a reduced

probability of predation is suggested to outweigh the cost of being spatially separated from the near-surface food, with a resulting reduced potential for daytime feeding (Hays, 2003). The less common twilight and reverse migration patterns have advantages as well, one of which could be to avoid other nocturnal migrators, as e.g. non-visually hunting invertebrate predators or simply competitors (Heywood, 1996; Ringelberg, 2009).

DVM is that much widespread and is found within practically all taxonomic groups, that it is generally assumed that in many

cases there must be a common underlying ultimate driving force (Pearre, 2003). Pioneering studies (Clarke, 1934; Eyden, 1923) hypothesized that migrators ascend into food-rich layers when hungry and descend after feeding, thus directly linking DVM to feeding. Likewise, Hardy (1953) and Stuart and Verheye (1991) suggested that carnivorous migrators, such as chaetognaths, might be simply following their herbivorous preys. However, in some cases, diel migration appears to have no link to feeding, e.g. when benthically feeding animals rise at night (as reported e.g. by Neverman and Wurtsbaugh, 1994). On

the other hand, theories of migration based only on light or temperature effects, as driving factors, might not fully explain this complex biological phenomenon and ignore individual behaviours and responses to the environment (Gibbons, 1993). Laboratory studies show that organisms kept constantly at dark, with similar in situ conditions, continue to maintain a damped DVM rhythm, with an evening ascent and a clear downward movement in the morning (Häfker et al., 2017). This suggests the importance of an endogenous circadian biochemical internal clock and might explain the midnight sinking, the sunrise ascent

(twilight migration) and DVM within the aphotic layer (van Haren and Compton, 2013). In fact, DVM is conditioned by a larger number of endogenous and exogenous factors (Ringelberg, 2009). Among endogenous factors there are sex, developmental stage, age, genotype, size, and internal rhythms (Richards et al., 1996), while exogenous factors include light, food availability, gravity, thermohaline characteristics (temperature, salinity, stratification), oxygen and hydrostatic pressure. Studying the diel vertical distributions of zooplanktonic biomass is essential to achieve a better understanding of the

functioning of pelagic ecosystems and the biological pump. By feeding near the surface at night, and then fasting at depth during the day, where it continues to defecate, respire and excrete, migrating zooplankton removes carbon and nitrogen from the surface layers and releases them at depth (Hays et al., 1997; Longhurst and Glen Harrison, 1989; Schnetzer and Steinberg, 2002). Vertical migrators (including both zooplankton and phytoplankton) play a relevant role in the vertical fluxes of matter and energy in the marine environment. The net direction of this flux is downward, although migrators are able to return

significant amounts of matter/energy upward, contributing to the effective recycling of nutrients within the euphotic zone (Pearre, 2003), thus supporting regenerated primary production.

Traditionally, DVM surveys are very time and labour intensive. Emerging technologies, such as acoustic techniques, can reduce this investment, greatly increasing the ability to decrypt the drivers, benefits for migrating organisms and total extent of vertical migrations. The Acoustic Doppler Current Profiler (ADCP) is a widespread instrument used to measure water



current profiles. Since the pioneering work of Flagg and Smith (1989) ADCPs are used to investigate zooplankton DVM and zooplankton biomass from measurements of vertical velocity and echo intensity (a measure of acoustic backscattered energy). In this study an ADCP, moored at about 400 m depth within the 443 m deep Corsica Channel (western Mediterranean Sea) between Corsica and Capraia islands (Fig. 1), was used to investigate the DVM of zooplankton and its biomass variations

along the water column from April 2014 to November 2016. The ADCP is part of a long-term fixed deployment and is used to measure water properties and currents (Schroeder et al., 2013), so the setting of the instrument was not originally thought for the application presented here. However, although the temporal and spatial resolutions are not in the optimal ranges, this method still provides a valuable insight on zooplankton DVM in the north-western Mediterranean Sea. The information derived by the ADCP is complemented by a morphological community analysis of in situ samples obtained with two net casts in the

same area in August 2015. CTDs performed from a ship during maintenance operations of the mooring and from a moored profiling system provided data to characterize the study site.

To better interpret the ADCP data it is essential to know which organisms are common in the zooplanktonic community of the Tyrrhenian and the Ligurian Seas. According to previous studies (Andersen et al., 1998; Pinca and Dallot, 1995; Sardou et al., 1996; Siokou-Frangou et al., 2010; Warren et al., 2004), copepods are the most important epipelagic mesozooplanktonic group

in terms of abundance and biomass in the Mediterranean Sea are (70% of the total zooplankton biomass during spring in the Ligurian Sea Sea; e.g. *Clausocalanus* spp., *Oithona* spp., *Oncaea* spp.). While Euphausiids, such as northern krill (*Meganyctiphanes norvegica),* siphonophores (e.g. *Chelophyes appendiculata*) and salps (e.g. *Salpa fusiformis* and *Thalia democratica*) are the most abundant macrozooplankton groups in the Ligurian Sea during spring. In their review on macrozooplankton and micronekton in the northwestern Mediterranean Sea, Andersen et al. (1998) and Sardou et al. (1996)

also mentioned hydromedusae (e.g. *Solmissus albescens*), pteropods (e.g. *Cavolina inflexa*), mysids (e.g. *Eucopia Unguiculata*), peneideae, and two species of micronektonic fish genus cyclothone. These authors also described the vertical migratory behaviour of north-western Mediterranean species, finding an intraspecific variability in some of them, that show a bimodal distribution of their population at two different depths, with consequent different migratory behaviour, originated by differences of size and season.

ADCPs have been used in previous studies to investigate DVM in the Mediterranean Sea, in particular in the Ligurian Sea (Tarling et al., 2001, and Bozzano et al., 2014), in the Ibiza Channel (Pinot and Jansá, 2001) and in the Cretan Sea (Potiris et al., 2018). Bozzano et al. (2014) used acoustic backscatter data from a moored ADCP to investigate zooplankton dynamics in the upper thermocline and to infer the composition of the community in the Ligurian Sea. In the same area, Tarling et al. (2001) combined data collected by a vessel-mounted ADCP and net samples and found that in September the dominant groups in the

first 500 m were euphausiids and pteropods during night, making inferences on the vertical migration velocities of these swarms as well. Pinot and Jansá (2001) studied DVM in the Ibiza channel, where they described light irradiance as the primary factor that controls DVM on the daily and seasonal basis. Potiris et al. (2018) studied the role of DVM for the functioning of the biological pump in the Cretan Sea, using a moored ADCP, CTD casts, net samples and other auxiliary information on environmental conditions, finding four different patterns of nocturnal DVM (divided by depth ranges). Other studies that



successfully used this technique were conducted in other parts of the world oceans, e.g. in the North Atlantic (Heywood, 1996; Jiang et al., 2007; van Haren, 2007; van Haren and Compton, 2013) and in the South Pacific (Valle-Levinson et al., 2014). Van Haren and Compton (2013) for instance investigated the link between the monthly lunar cycle and the DVM of deep planktonic organisms and pointed out the importance of the biochemical internal clock, while Valle-Levinson et al. (2014)

found that twilight migration was predominant within Chilean fjords and was strongly influenced by the depth of the depth of the pycnocline. Most of these studies denote that acoustic data are more qualitative than quantitative, because attempts to calibration sound backscatter and biomass from net samples are complex and not yet satisfactory (Flagg and Smith, 1989; Pinot and Jansá, 2001).

Vertical velocity data show when zooplankton moves and in which direction, while data of acoustic backscattered energy

allow to determine how much zooplankton is present at a certain depth range and a certain time. In this study it is investigated how both parameters change at different temporal scales, from daily to seasonal, and at different depth ranges. Additional data (CTD casts, net samples, satellite data, sunrise-sunset hours, moon phases) are used to identify the drivers of zooplankton migration in the Corsica Channel, the zooplanktonic groups that can be found in the area, what kind of migration they do perform and how biomass varies along the water column and in time.

The paper is organised as follows. First, the study area is described, based on previous knowledge and on a literature review, then, in section 3, the ADCP settings and quality control procedure are described, along with the explanation on how to compute the mean volume backscatter strength from the ADCP data. Data collections by means of CTD casts, moored profiling systems, net samples and additional systems and methods are described in the rest of section 3. The presentation of the results and their discussion (section 4) starts with the characterization of the water column in the Corsica Channel (thermohaline

properties, stratification, oxygenation, depth of the chlorophyll maximum) and the description of the acoustic backscatter and vertical velocities on the daily and the seasonal scale. The zooplankton community composition in summer 2015 is then described afterwards and put in relation to the acoustic observations of the same period. The section is completed by a lagged correlation analysis of the backscatter data and a time series of primary production in the area, to look for the timing of primary production blooms vs secondary production blooms. Finally, the conclusions are drawn at the end of the paper.

## 2 Study Area

The Corsica Channel separates Corsica and Italy and is the only (narrow) connection between the Tyrrhenian and the Ligurian Seas. Two water masses flow through this channel: the Atlantic Water (AW) in the upper layer and the Intermediate Water (IW) between 150 m and the bottom (maximum depth of about 450 m). The IW is the saltiest and warmest water mass of the whole Mediterranean Sea and originates in the eastern Mediterranean Sea; the AW comes from the Atlantic Ocean, crossing

the Gibraltar strait, flowing into the Mediterranean Sea. While moving eastward above the IW, the AW is continuously modified by the interaction with the atmosphere and the underneath water masses, becoming gradually saltier and denser (Millot and Taupier-Letage, 2005). Both water masses enter the Tyrrhenian Sea from the south and then follow a cyclonic





circulation along the Italian peninsula. When reaching the northernmost Tyrrhenian, parts of AW and IW cross the Corsica Channel (as the Eastern Corsica Current, ECC), where the mooring is located (Fig. 1), reaching the Ligurian Sea. The IW flows through the channel only in its deepest part, located between the islands of Corsica and Capraia. The flow is generally northward, stronger between winter and late spring (mean velocity 0.15-0.2 cm/s), weaker during summer until late autumn

(mean velocity 0.05-0.1 cm/s). This pattern undergoes noticeable variations of intensity and duration mostly in the stronger flow period (Astraldi and Gasparini, 1992). To the north of Corsica, the ECC merges with the Western Corsica Current (WCC). The resulting current proceeds northward and then westward becoming the so-called Northern Current, a geostrophic frontal system along the continental slope, dividing coastal waters from denser waters of the central Ligurian Sea (Millot and Taupier-Letage, 2005).

The Mediterranean, as a whole, is considered an oligotrophic sea. The north-western Mediterranean (e.g., the Ligurian Sea), however, exhibits large areas of high chlorophyll values thanks to the upwelling in the central part of the basin induced by the cyclonic circulation, providing conditions for enhanced primary productivity, and a classical spring bloom. On the other hand, the Tyrrhenian Sea only has intermittent spring blooms, i.e. characterized by significant interannual variability (D'Ortenzio and D'Alcalà, 2009). The region of the Corsica Channel has intermediate characteristics between these two adjacent

biogeographic regions.

## 3 Materials and Methods

### 3.1 ADCP settings and data quality control

The operating principle of ADCP is based on sound back-scattering by particles (as sediments, organisms or bubbles) suspended in the water. The instrument emits acoustic impulses, with known frequency and receives the echoes, with a shifted

frequency. The frequency shift is directly proportional to how fast particles move (Doppler effect) and is used to infer the velocity and direction of passive particles suspended along the water column (Teledyne RD Instruments, 2011). The basic assumption is that the particles are passively carried by water masses, and that they move together at the same speed. It is not possible to determine exactly how much sound reflection is due to zooplankton, since the acoustic waves are reflected by all objects of the size of about ¼ wavelength of the acoustic impulses (Thomson and Emery, 2014). If we consider the speed of

sound in seawater around 1475 m/s and the ADCP working frequency of 76.8 kHz, the wavelength is about 1.9 cm, so objects greater than 0.48 cm reflect sound, while objects smaller than this scatter the sound. However, since swarms of zooplankton tend to aggregate at specific depths, also smaller organisms can be easily detected because acoustic backscatter strength is proportional to the density distribution of organisms (Iida et al., 1996). In zooplankton DVM studies, usually two important assumptions are made: vertical velocity, detected under general oceanic conditions by an ADCP, is due principally to

zooplankton motion, with negligible upwelling and downwelling phenomena (Heywood, 1996) and sound backscatter is due principally to zooplanktonic biomass. Sound back-scattering is influenced by organism shape, orientation (Chu et al., 1992) and consistency, e.g. organisms made up mostly of protoplasm do not backscatter the acoustic signal proportionally to their





size (Flagg and Smith, 1989). Therefore, the data of zooplankton biomass and vertical velocity obtained by ADCP are more qualitative than quantitative.

Data of echo intensity and vertical velocity (W) were collected with an RDI WH Long Ranger 76.8 kHz ADCP, an instrument that is used in a long-term deployment and has a wide profiling range. The ADCP has four beams, which emit sound signals

and receive echoes. These are put at 90° azimuthal increments each other and pointing at 20° to the instrument axis. The four beams work as transducers converting sound signals in electrical signals. The RDI WH Long Ranger is upward looking (the beams emit sound towards the surface) and is moored at about 400 m depth, near the bottom (which is at 443 m depth) of the Corsica Channel, between Corsica and Capraia islands (position 43.03° E, 9.68° N). The time series used for this study spans from April 5th 2014 to November 26th 2016. During the collecting period, the ADCP has been recovered 6 times for

maintenance, therefore there are six interruptions (generally < 24 h) in the time series. The time series of vertical velocity and echo intensity were collected with a temporal resolution of 2 hours, an ensemble value resulting of 45 or 60 pings average (which means a sound pulse every 2.4 or 2 minutes, depending on the deployment configuration), and a vertical spatial resolution of 16 meters, which is the length of the depth cells (or bins) in which the vertical profile is subdivided. The blanking length, where the instrument does not measure, is 7.04 m above the transducer. All details of the ADCP setting during the 7

deployments are listed in Table 1.

It is noteworthy that a higher temporal resolution would have given a better estimation of patterns of vertical migration: during the averaging period of 2 hours some of the 45 or 60 pings might have intercepted organisms while they were moving, while other pings might have intercepted organisms when they were almost still at their comfortable depth. The spatial and temporal resolution was set to allow the mooring to be in place for about 6 months without compromising battery endurance and space

on the internal memory.

While echo intensity data need additional processing (see section 3.2), W data did not need further handling, except for some data selection criteria and quality control considerations to discard the low-quality data (this was applied also to backscatter data). Given that the total bin number was set to 28 and considering the blanking length plus the bin size of 16 m (D), there were at least four bins above the sea surface, which were discarded. Also, the first bin, closest to the transducers, is not used

because it may record erroneous data due to the time taken for transient acoustic waves to decay (Lane et al., 1999). Moreover, R, the slant range, i.e. the range of relevant scattering layers along each beam (determined by Eq. (2) following Deines, 1999), must satisfy $R_{min}> \pi R_0/4$ and $R_{max}< H cos\theta$ (Teledyne RD Instruments, 2011). With $R_0$, the Rayleigh distance, being 1.3 m for this ADCP model, all values from the 20th bin upwards, for deployments #1, #2, #3, #5 and #6, exceeded $R_{max}$, and were discarded.  Only in the 4th deployment the values detected in the 20th bin was not discarded, because the ADCP was at 411

meters in this deployment, the deepest one (see table 1). Thus, in Eq. (2) N maximum value is equal to 20 for the 4th deployment and 19 for all other deployments. To avoid tilt error, pitch and roll of the instrument must not exceed 15°, and the data collected when pitch and roll were higher than 15° have been discarded as well (Teledyne RD Instruments, 2011). Only few data were discarded due to this criterion, mainly in late winter and early spring, because of the strong currents that occur in this period of the year (Astraldi and Gasparini, 1992), which can cause the inclination of the entire mooring line. A last data selection



criterion was the Percent Good (PG) that had to be greater than 90%. PG is a measure of the percentage of pings accepted to obtain the ensemble value of vertical velocity or echo intensity. Given all these constraints, ADCP gives information on DVM in a layer between about 70 m and 390 m. All considerations that will be done in the following need to take into account that there is a lack of information concerning biomass and migration in the very surface layer and in the 50 m above the bottom.

**3.2 Estimation of the Mean Volume Backscatter Strength**

To express the measured quantities in sound backscattered energy instead of echo intensity (which is measured in counts), first the Mean Volume Backscatter Strength (MVBS), measured in dB re $4\pi$/m, is calculated following Eq. (1), as described in Deines (1999):

$$MVBS = C + 10log_{10}\left[(T_x + 273.16)R^2\right] - L_{DBM} - P_{DBM} + 2\alpha R + K_c(E - E_r),  \quad (1)$$

where C is a constant factor specific of the ADCP model used (dB); $T_x$ is the temperature detected at the transducer (°C); R is the slant rage (m) as defined by Eq. (2); $L_{DBM}$ is the $10log_{10}$ of the transmit pulse length (m), which is specific for each deployment; $P_{DBM}$ is the $10log_{10}$ of the transmit power, specific for this ADCP model (24 W). The term $2\alpha R$ contains the coefficient of sound absorption in seawater (Fisher and Simmons, 1977) at the specific bin depth $\alpha$ (dB/m), computed by Eq. (3) following Deines (1999), $\alpha$ depends on the frequency of the sound pulse in Hz (76800 Hz in this case), temperature in °C 15 ($T_x$) and pressure in atmosphere. $K_c$ converts counts in decibel and is defined by Eq. (4) (Heywood, 1996). E is echo intensity (counts) calculated by averaging echo intensity detected by the four beams, while $E_r$ is the noise value, i.e. the echo intensity detected by the instrument when there is no signal (50 counts in this case). So, R is calculated as follows:

$$R = \frac{B + \frac{(L+D)}{2+[(N-1)\times}D] + (\frac{D}{4})}{cos\,\theta} \times \frac{c}{c_0},  \quad (2)$$

where B is the blank distance from transducers to the first bin (7.04 m for all deployments); L is the transmit pulse length (m);
D is the cell, or bin, length (m); N is the number of the cell (bin number); the angle $\theta$, in degrees, is the inclination of each beam respect to the vertical axis of the instrument (20°); c is the sound velocity (m/s) for each bin (computed following IOC/SCOR/IAPSO, 2010) which depends on salinity (a nominal value of 38 has been used), temperature in °C and pressure in dbar; $c_0$ is the sound speed in seawater used by the ADCP (1475.1 m/s). The equation to compute the term $2\alpha R$ is Eq. (3):

$$2\alpha R = \frac{2\alpha_p B}{cos\,(20)} + \sum_{n=1}^{b} + \alpha_n,  \quad (3)$$

where $\alpha_p$ (dB/m) is the sound absorption at the depth of the ADCP; b is the last bin number; $\alpha_n = 2\alpha D/cos(20)$ is the sound absorption for each cell. The formula to compute $K_c$, that appears on the right-hand-side of Eq. (1), is given in Eq. (4):

$$K_c = \frac{127.3}{T_z + 273},  \quad (4)$$

All parameters are summarized in Table 1.



### 3.3 CTD data

During servicing, between one deployment and the following one (see dates in Table 1), CTD casts are regularly performed, from surface to bottom. These 6-monthly data are useful to provide information on the stratification and the depth of the chlorophyll maximum along the duration of the experiment. Each time pressure, conductivity, temperature, dissolved oxygen

concentration and chlorophyll fluorescence were measured with a CTD-rosette system consisting of a CTD SBE 911 plus, a Wetlab fluorescence sensor, and a General Oceanics rosette. The CTD probes were calibrated before and after each cruise (dissolved oxygen and salinity also during each cruise). Maintenance operations and CTD casts were done from the Italian vessels *R/V Urania* and *R/V Minerva Uno*.

In addition, a profiling buoy system for real time data transmission has been mounted on the mooring from November 28[th]

2014 to March 20[th] 2015. The system is composed of two units: (i) a profiling buoy, carrying a CTD sensor (with temperature, salinity, oxygen and chlorophyll fluorescence sensors), Iridium antenna, and (ii) an underwater winch. Both units are provided with acoustic remote transceivers to communicate with each other, and with a deck unit. The profiling system is moored at 190 m depth on the mooring line, and it has been set to perform an upcast CTD profile from 190 m to surface once a day. Conversely to CTD casts, which are only snapshots of the thermohaline conditions at a specific day and time, these data gives

a daily information on the whole upper layer for several months. A previous deployment in 2013 is extensively described in Aracri et al. (2016).

### 3.4 Zooplankton net samples

The backscatter strength and vertical velocities data collected by ADCP were integrated by data on zooplankton community composition, obtained from two net samples retrieved in the Corsica Channel. One net tow was done at the mooring location

(Sample #1, August 24[th] 2015 at 8:37 UTC, bottom depth 443 m), while the mooring was recovered for maintenance, and the second one about 6.5 km to the west (Sample #2, 43.03° N, 9.60° E, August 24[th] 2015 at 10 UTC, bottom depth 234 m), from the Italian vessel *R/V Minerva Uno*. As the sampling net did not reach the bottom (it remains 10-15 meters above it), some organism might not be sampled if they stay in the deepest layer, close to the bottom, a common behaviour especially during the day (Vinogradov, 1997). Indeed, populations of many pelagic species extent into the hyperbenthic and benthopelagic

environments within a few meters from the seafloor, where there may be significant accumulation of biomass during the day in specific seasons (Mauchline, 1998 and references therein). The two stations were sampled for the taxonomic and quantitative characterization of mesozooplankton communities. Samples were collected by vertical hauls, almost from the bottom to the surface, using a standard Indian Ocean net equipped with flowmeters for filtered-volume calculation (1.13 m diameter and 200 µm mesh size) and preserved with borax-buffered formaldehyde. Taxonomic and quantitative zooplankton determinations

were performed using a Zeiss stereomicroscope at the lowest possible taxonomic level (species level for copepods and cladocerans) on a representative subsample, while the total samples were analysed for rare species determination.



### 3.5 Additional ancillary data and statistical methods

Additional environmental parameters were used for this study, to investigate a potential correlation with vertical migration and the amount of biomass in the Corsica Channel and to explain what drives them. These parameters are sunrise and sunset time (using the script *suncycle*, downloaded from here:  http://mooring.ucsd.edu/software/matlab/doc/toolbox/geo/suncycle.html);

surface Chlorophyll *a* concentration (Chl *a* in mg m$^{-3}$, 1 km resolution, 8-days averages) in the area of the mooring (downloaded for the domain latitude=43.0097°N, 9.4°E<longitude<9.8°E), computed via regional algorithms (Volpe et al., 2007) and retrieved from the COPERNICUS Marine Environment Monitoring Service, or CMEMS  (product name "OCEANCOLOUR_MED_CHL_L4_REP_OBSERVATIONS_009_078",         downloaded         from         here: http://marine.copernicus.eu/services-portfolio/access-to-products/); the moon phases to estimate the potential effect of

moonlight on vertical migration patterns.

Two statistical analyses were applied on the MVBS and W datasets, a spectral analysis using the fast Fourier transform (FFT) and a lagged correlation analysis. FFT was applied to identify the most relevant oscillations in the vertical migration patterns, observing the peaks with the highest amplitude at both high and low frequencies. Low frequencies peaks were determined after applying a low pass filter (frequencies < 5x10$^{-7}$ Hz, that is approximately 23 days). The lagged correlation analysis

between MVBS and Chl *a* was done to verify if in this area the primary production results to be a relevant driver for secondary production (for which MVBS is considered to be a proxy).

### 4 Results and Discussion

The data collected by the ADCP are used to define the temporal and spatial variability of zooplankton DVM and biomass distribution patterns during the investigated period. Additional environmental data are derived from CTD casts and satellite in

order to understand what drives zooplankton behaviour and blooms, while the taxonomic analysis of the zooplankton net samples is used to describe the community structure. In the following sections, results and their discussion are divided in four parts, starting with the description of the water column in terms of thermohaline characteristics and stratification in different seasons. Daily and seasonal cycles of acoustic backscatter intensity and vertical velocities are then discussed, while the third part is devoted to the discussion of the biological net sample data, that allow to gain a picture of which groups of organisms

were present within the Corsica Channel during late August 2015. To conclude, the temporal pattern of phytoplanktonic blooming vs non-blooming periods (estimated by using Chl *a* concentration from satellite data) is compared to secondary production patterns (considering the integrated MVBS a proxy for zooplankton biomass in the water column) in the Corsica Channel.

### 4.1 Thermohaline characteristics within the channel

Seasonal variability of thermohaline characteristics in the area evidences marked differences between the stratified water column in summer and unstratified water column in winter. CTD data collected during servicing allow to investigate this



behaviour (Fig. 2a-2d). CTD casts in March 2015 and 2016 represents the ending phase of winter conditions, with homogeneously stratified temperature vertical profiles and relatively higher level of dissolved oxygen at depth, as a results of wintertime open ocean convection. Chlorophyll fluorescence shows maxima in the surface layer (Deep Chlorophyll Maximum, or DCM, at 20-30 m) approaching the spring bloom period and a weak secondary relative maximum in correspondence of an

oxygen maximum at depth (200 m in March 2015 and 250 m in March 2016), possibly relict phytoplanktonic populations transported downward by vertical mixing or photosynthetic picoplankton able to use the wavelengths and low light levels that are characteristics of this depth. In summer the water column is well stratified, with the development of a sharp thermocline in the uppermost 20-40 m, lower surface oxygen contents, and a DCM at about 100 m (Fig. 2d). In fall, the surface layer undergoes a progressive cooling toward winter, the thermocline being at about 50-60 m and the DCM becoming weaker and

shallower (60 m). Salinity below 200 m (except in winter when this interface is deeper) is generally homogeneous and this level can be considered the transition between the AW above and the IW below (Fig. 2b).

The evolution through winter can be followed by means of the daily data time series collected by the moored profiler (profiling range between 0 and 180 m) that was in place from November 28[th] 2014 to March 20[th] 2015 (Fig. 2d-2e). Progressive cooling of the water column continues till late January (Fig. 2d), when fully mixed conditions are eventually met. Conversely, dissolved

oxygen (Fig. 2f) as well as chlorophyll fluorescence (Fig. 2g) gradually increase in the whole upper layer while approaching spring season.

## 4.2 Acoustic backscatter and vertical velocity

Vertical velocities along the water column and backscatter strength are analysed to identify zooplankton motions and biomass variations, and to characterize different migratory behaviours of different zooplanktonic migrator groups. To this aim, lacking

the necessary data for a proper calibration, MVBS is considered as an indirect proxy of zooplankton biomass. Since with a sampling period of 2 hours W values result very low and does not represent the actual velocity of these organisms, it is nevertheless used to provide insights on the net direction of motion (up or down) according to the hour of the day, season and depth range.

The data collected over the entire period of the seven deployments (April 2014-November 2016) are shown in Figure 3. MVBS

is the Mean Volume Backscatter Strength computed with Eq. (1) for each bin, while its anomalies (Fig. 3a) are obtained by subtracting from each MVBS profile the average MVBS profile of the entire period. All considerations that follow do take into account that there is a lack of information concerning biomass and migration in the very surface layer and in the 50 m above the bottom.

MVBS anomalies (Fig. 3a) clearly present periodic oscillations, with notably higher than average values between

November/December and April/May, a bloom that involves the whole investigated water column. High surface values, associated to low deep values are observed outside the blooming periods. Since MVBS is a proxy of secondary production, the observed variability is probably linked to the primary production seasonality as well as to the alternation of stratified and



mixing periods, as described earlier (Estrada et al. 1985). The peaks of the blooming period in 2015 and in 2016 are slightly different, with 2015 presenting a prolonged and more intense increase in MVBS, as compared to 2016.

Less evident in Fig. 3a, there is also a pronounced daily cycle, as expected. To show its features more in detail, Fig.3b-3g represent the temporal evolution of MVBS and W at selected depths (within the surface, intermediate and deep layers) as a

function of the hour of the day (UTC), and with the times of sunset and sunrise (that change seasonally) superimposed.

In the surface layer (bin centred at 97 m) MVBS is clearly higher during the night and lower during the day (Fig. 3b). Summer 2016 behaves differently compared to summer 2015, with very high values persisting night and day (June - July 2016). During the blooming periods, MVBS peaks from 2 to 4 hours before sunset, a pattern that is consistent with the presence of seasonal twilight migrating organisms. W in the surface layer (Fig. 3c) is clearly directed upward (positive W) at sunset and downward

(negative W) before dawn, during the whole duration of deployment. This is consistent with the classical picture of DVM. In February - March (2015 and 2016) there are very strong positive values persisting night and day.

In the intermediate layer (bin centred at 209 m) MVBS has a more pronounced daily pattern than in the surface layer (Fig. 3d), with nocturnal high backscatter strength and diurnal MVBS minima. The summer 2016 persistent high values found in the surface layer are absent at mid-depth. Also here the MVBS starts to rise from 2 to 4 hours before sunset, especially during the

blooming period, as observed in the layer above. The patterns of descent and ascent (Fig. 3e) are clearly observed throughout the whole period and follow closely the seasonality of sunrise and sunset times. Downward velocities at sunrise are much stronger than in the surface layer and also than the upward velocities at sunset. In summer (2015 and 2016, less in 2014) there is a strong upward motion just after sunrise, which is coherent with twilight or reverse migration patterns.

In the deep layer (bin centred at 353 m) MVBS is very high during the whole experiment (Fig. 3f), with small differences

between day and night. We discarded the possibility of this layer being a nepheloid layer, after investigating historical turbidity data at the same location (high turbidity levels were not found at depths shallower than 410-420 m, which is below the depth of the ADCP). Overall, it appears that daily values are slightly higher than nocturnal values of MVBS, suggesting that some organisms migrate from higher levels to this depth during the day. However, Fig. 3g suggests that in the deep layers the migration is much lower. It is likely that this layer is occupied by non-migrating organisms or organisms that have a reduced

migration. During the blooming period in winter-spring, MVBS reaches the highest levels, with no difference between day and night, and with 2015 showing a more intense peak than 2016. At this depth, W (Fig. 3g) is not clearly correlated with sunlight, with prevalent negative velocities occurring almost at all times. Downward motions are stronger in 2016 from late winter to spring, in summer 2014 and 2015 during night and in the hours before and after sunrise. Upward motions are very weakly correlated with sunset and slightly increase from noon to sunset and during the 2015 blooming period.

To investigate more in detail the seasonal variability of MVBS along the water column, as well as the different patterns of MVBS and W during blooming and non-blooming periods, the seasonal cycle and the evolution of DVM parameters as a function of the hour of the day and depth are shown in Fig. 4.

In particular it is observed that the highest values of MVBS, integrated over the whole investigated water column, occur between November/December and April/May (Fig. 3a and 4a), which corresponds to the zooplanktonic blooming period, and



with a peak that involves the whole water column in February-March. The associated standard deviation (Fig. 4b) shows that the blooming period is also the one with less variability. During the rest of the year (the non-blooming period), MVBS is very low, especially at mid-depth (between 150 and 300 m), while it presents a marked interannual variability, as evidenced by the standard deviation (particularly high between 200 m and 330 m, from June to October). Such MVBS seasonal pattern is likely

to be the response of zooplankton to both the different thermohaline conditions of the water column (MVBS increases when stratification is weaker and the thermocline is almost absent, see section 4.1) and the seasonality of phytoplankton blooms and DCM position (see section 4.1 and the following 4.4 for details). During summer-autumn, when stratification is stronger and the DCM is deeper (Fig. 2d), MVBS maxima are split into two layers (Fig. 4a), a shallower one and a very deep one, which is likely to be due to the presence of two zooplanktonic communities with different depth-based habitat preferences (as found

also by Heywood, 1996, and Pinot and Jansà, 2001). This is a consistent pattern, as denoted by the mostly low standard deviations in these two layers during the non-blooming period (Fig. 4b). Since the ADCP measurements miss the first tens of meters of the water column, the summertime increase of MVBS at 70-100 m might be also a consequence of a cyclic summer descent (due to the increase of irradiance) of a group of epi-zooplanktonic organisms, that during the rest of the year finds food and optimal light and temperature conditions in more superficial waters, which the ADCP data were not able to detect. It is

known that in the western Mediterranean during summer the zooplankton biomass maximum at daytime is concentrated around the same depth as the DCM (in the range from 70 to 90 m, which is close to the upper limit of the present observations), while at night this maximum raises up to less than 20 m (Alcaraz, 1985).

To characterize the different DVM patterns during no-blooming and blooming periods, MVBS and W variations as a function of depth and of the hour of the day are shown as average values between May and November (defined as the no-blooming

period, exact dates were defined based on the integrated MVBS values falling below a certain threshold) and between December and April (defined as the blooming period, exact dates were defined based on the integrated MVBS values exceeding the same threshold) in Fig. 4c-4d and in Fig. 4e-4f, respectively. At a first analysis of these figures, sunlight is easily identifiable as the most important driver of diel vertical migration both during no-blooming and blooming periods.

During the no-blooming period MVBS shows a bimodal distribution, with high biomass levels being evident both in the upper

layer (above 120 m) and in the bottom layer (below 330 m), and very low levels at mid-depth (Fig. 4c), a feature that was evident also in the seasonal full-depth analysis in Fig. 4a. In the course of the day the mid-depth minimum becomes thicker, expanding mainly towards the deeper levels (Fig. 4c): although thinner, the minimum layer persists also during night, occupying the depth range of 150-250 m, as opposed to the 120-350 m range occupied during day (with maximum thickness at midday). In the upper layer MVBS is higher during night than during day, while at depth it maintains approximately a

constant level, with only a slight increase during day. Vertical motion is directed downward along the whole water column during night, with a maximal intensity at dawn (4-6 UTC) and bidirectional during the day, with a maximum upward intensity at dusk (16-18 UTC) above 300 m (Fig. 4d).

During the blooming period the bimodal distribution of MVBS is weaker (Fig. 4e), with biomass in the upper layers exhibiting lower levels compared to both the deep layer and to the upper layer during the non-blooming period (Fig. 4c). However, it has





to be considered that no data are available for the most superficial bins for this period (as a consequence of the quality control applied to the raw data, see section 3.1), so it remains questionable whether more biomass is found above these levels or not. During the blooming period the MVBS minimum layer is thinner and resides at shallower depths if compared to the non-blooming period (80-270 m instead of 120-350 m depth range). In addition, the day-to-night differences of this layer are less

pronounced during this period (Fig. 4e). Vertical motion during the blooming period (Fig. 4f) is directed downward at 6-8 UTC (a bit later because of later sunrise times during the blooming period), while the upward migration occurs mostly at 16-18 UTC, and is more intense than during the non-blooming period (Fig. 4d). Thus, it appears that in the investigated water column DVM is intensified during the blooming period and that biomass in the upper layers is relatively lower than during the non-blooming period.

These outcomes are consistent with the hypothesis (Hardy and Gunther, 1935; Huggett and Richardson, 2000) that when food availability is high (as occurs during phytoplankton blooms, which will be discussed in section 4.4), the migration is intensified, because herbivorous zooplankton feeds enough during the night to stand the costs of not-feeding during the day by descending in deeper layers in order to hide from visual predators. In contrast, when food availability is scarce (non-blooming periods), those organisms have to take the risk of predation by staying in surface layers during the day to compensate the shortage in

food sources. However, it needs to be taken into account that the observed differences during the two periods may also be explained by a community shift and other environmental factors, e.g. stratification, thermocline depth and position of the DCM. Indeed, according to Angel (1968) and Ringelberg (2009) a strong thermocline has a negative effect on vertical migration, which implies that the bimodal distribution and the reduced vertical migration observed during the non-blooming period can also be attributed to the strong thermoclines that develop during late-spring/summer (Fig. 2a and section 4.1).

As has been described and depicted in Fig.3b-3g and Fig4-4f, nocturnal migration with a 24 hours cycle (a circadian cycle conditioned by sunlight) is the most evident type of migration in the study area. Yet some other migrating cycles could be hidden. For instance, from Fig. 4d and 4f, it appears that there is a strong descent after sunset, at 20-22 UTC during the non-blooming period (less strong at 18-20 UTC in the blooming period), which could be identified as reverse migration. During both periods, an upward motion is evident after sunrise, a feature that is characteristic of twilight and reverse migration. In

order to identify other migration patterns, a fast Fourier transform (FFT) is applied to the dataset of MVBS and W (Fig. 5a and 5b, respectively). The spectral analysis is applied also to the low-pass filtered time series to identify lower frequencies signals (Fig. 5c and 5d, respectively for MVBS and W).

It is well evident that MVBS and W have the same peak frequencies at periods lower than 1 day, although with differences in amplitude (Fig. 5a-5b). The most evident peak is at 24 hours, as expected by the prevalent nocturnal DVM pattern, as well as

by the less frequent reverse migration. The amplitude of the peak is the highest at mid-depth (bin centred at 209 m), while it decreases both upward and downward (being minimum within the bin centred at 353 m). This difference between layers has already been observed while discussing Fig. 3b-3d-3f. At 12 hours there is a prominent peak at 97 m and 209 m (hardly visible at 353 m), both for MVBS and W, which might be due to some groups that do reverse migration: although taken singularly this migratory behaviour has a 24 hours cycle as well, and occurs at sunset (descent) and sunrise (ascent) as does nocturnal




migration, if reverse and nocturnal migration both occur this can produce a signal at 12 hours (the time lapse between two consecutive ascending events and two consecutive descending events is around 12 hours). The fact that the 12 hours peaks are less intense than the 24 hours peaks suggests that reverse migration does not take places all over the year or does not involve a large number of organisms. This would also explain why reverse migration was not evident in Fig. 3b-g. Other authors

(Bozzano et al., 2014; Picco et al., 2016) consider the 12 hours peak due to twilight migration. The other peaks, at 8 hours, 6 hours (both very strong at 209 m, almost absent at 353 m) and 4.75 hours (not visible at 353 m), in both MVBS and W spectra, could be due to different groups performing patterns of twilight migration (ascent at dusk and sunrise, descent at midnight and immediately after sunrise), with 4.75 hours being consistent with the mean time lapse between midnight descent and sunrise descent. Indeed, in Fig. 3d (W during the no-blooming period) it is possible to see ascending motions right after the descent at

sunrise, followed by upward velocities at sunset, i.e. 8 hours later on average. The low amplitude of these peaks again suggests that also twilight migration does not take place all over the year or does not involve a large number of organisms.

Notable low-frequency peaks in the low-pass filtered MVBS and W time series (Fig. 5c-5d) are at 28-30 days, which might indicate a cycle connected with moon phases, and thus with the alternation of strong and weak moonlight during night. The 80-96 days peaks (visible at all three depths, but less intense at 353 m) are clearly related to the alternation of the four seasons,

while the 160-193 days peaks reflect the broader periods of blooming and non-blooming. Indeed, there are so many differences between them, as seen earlier, and they correlated with water column properties (section 4.1). The 322 days peak (not visible at 97 m for MVBS) is almost one-year period and since our time series is just 2 years and 8 months long, this simply corresponds to the mean time lapse between two consecutive spring maxima (and summer minima) at the three selected depths.

### 4.3 Zooplanktonic community composition in summer 2015 and associated DVM patterns

In order to describe the zooplankton community, two net samples collected in the study area in August 2015 are discussed in detail in the following, keeping in mind that these samples cannot give insights into the temporal variability of the community (they rather give a snapshot of it during day and in summer) and that the vertical distribution is not resolved, being the samples collected by integrated vertical tows.

In the two stations, by far the most abundant group were the copepods with 83% ±0.4 of the total community, followed by

other taxa, mainly represented by appendicularians and chaetognaths, with 13% ±2.8 and then by cladocerans with 4% ±3.2 (Fig. 6a).

Both stations showed a very similar community dominated by few species, mainly belonging to epipelagic copepods, with the two most abundant genera, i.e. *Clausocalanus* spp. and *Oithona* spp., accounting for more than 50% of the total abundance (Table 2). In the more western and slightly shallower station (sample #2), the abundance of cladocerans was higher compared

to the station at the mooring location (sample #1), as is evident from Table 2. The community is essentially composed by organisms that do not migrate significantly, which is consistent with the reduced migration during summer detected by ADCP (Fig. 4d). Most organisms found in the samples were smaller than the size detection limit in this configuration (0.48 cm), therefore the ADCP detects them only in high-density aggregations.





To explore more in detail the DVM patterns that involve the sampled community, the evolution of MVBS anomalies around the time of the zooplankton sampling (± 15 days) is shown in Fig. 6b. Around new moon MVBS shows lower levels than around full moon, especially in the more superficial bins, which is consistent with the different light conditions during night. An evident pattern visible in Fig. 6b is the descent during the day, between 150 and 250 m, performed by organisms of shallow layers. The alternation between night and day is clearly visible in Fig. 6b, as well as the presence of some groups performing migrations throughout the whole investigated water column (about 100-300 m): these could be macrozooplanktonic organisms, as suggested by, e.g. Pinot and Jansà (2001) and Heywood (1996).

As described earlier (see Fig. 4a), August is a period of low MVBS anomalies, with the exception of the most superficial bins. The increase of biomass in the layer between 60 m and 80 m, as shown in fig. 2d, can be explained by the summer deepening of the DCM, which is accompanied by a descent of the zooplankton maximum (i.e., from the very surface layer, outside the range of the ADCP, down to 60-80 m depth). This is consistent with the behaviour of the sampled community (e.g., *Clausocalanus* spp. and *Oithona* spp.).

**4.4 Primary and secondary production**

To understand how primary production drives the seasonal cycle of secondary production (Fig. 4a) in the Corsica Channel, in Fig. 7a a comparison is made between the temporal evolution of the 8-days Chl *a* average in the area of the mooring location and the 8-days averages of the integrated MVBS anomalies (obtained by summing up, along the vertical, the MVBS anomalies of each bin) of the whole investigated water column, of the shallow layer (73-201 m) and of the deep layer (201-378 m) during the whole deployment period.

It is clearly visible that MVBS anomalies and Chl *a* have a similar temporal evolution, with only slight differences in the timing of seasonal peaks: in late November 2014 a small Chl *a* peak and a contemporary peak of MVBS occurred; between early February and March 2015 an important zooplankton bloom follows a Chl *a* peak in January 2015 and occurred while Chl *a* again peaked in March 2015; in summer 2015 there were three little MVBS peaks that are absent in summers 2014 and 2016, which explains the high standard deviation during summer shown in Fig. 4b; finally in late winter 2016 Chl *a* reached its annual maximum, which was accompanied by a bloom in secondary production.

To further investigate the primary and secondary production blooms, in Fig. 7b the results of a lagged correlation analysis between MVBS (total, shallow and deep) and Chl *a* is shown. There is no lag when comparing the total and deep MVBS with Chl *a* (that well correlate), indicating that the data series co-vary with the same timing (on the 8-days window). When considering only shallow MVBS it results that the peaks in primary production precede the peaks in secondary production by about 3 weeks in the Corsica Channel. This pattern is what is expected from previous knowledge, since generally after a month since the surface primary production bloom there is a bloom in zooplankton biomass (Truscott and Brindley, 1994). The absence of any temporal lag for total and deep MVBS vs. Chl *a* in the Corsica Channel is somewhat unexpected, but it is necessary to keep in mind that the temporal resolution of the Chl *a* field from satellite is 8 days and that it is a surface value and not an integrated value of the primary production within the whole euphotic layer. Furthermore, the MVBS data do not





reach the very surface layer and the very bottom layer, where some zooplankton organisms might concentrate or peak with different timings. In addition, according to Madin et al. (2001) if the bulk of zooplankton within a water column is composed by vertical migrators, its growth dynamics are not necessarily only coupled to surface primary production. Correlation between MVBS and Chl *a* in the Ligurian Sea has been investigated by Warren et al. (2004) and McGehee et al. (2004), who observed

that small (large) zooplankton and Chl *a* was negatively (not) correlated. Zooplankton biomass and distribution are strongly related to hydrodynamic processes (Champalbert, 1996). Due to the mainly northward current and the role of hydrodynamic processes in Corsica Channel, we consider that the study area is strongly influenced by the biological processes that occur upstream, i.e. in the northern Tyrrhenian sea, an oligotrophic sea that comprises neritic waters where biomass might be higher and blooms can occur earlier as compared to oceanic waters. Phytoplanktonic blooms in the neritic areas of the northern

Tyrrhenian and the Ligurian Seas occur in late winter early spring, which corresponds to what can be seen in Fig. 7a (Marchese et al., 2015). Strong currents could be responsible of changes in the amount of zooplankton in the water column during the blooming period (when currents are stronger, see section 2), and organisms could have been brought in the region by lateral advection, and not be supported by local phytoplankton blooms.

**5 Conclusion**

DVM, one of the most important survival strategies adopted by zooplankton, has been investigated in the Corsica Channel, connecting the Tyrrhenian and the Ligurian Seas (western Mediterranean). An analysis of acoustic backscatter (MVBS) and vertical velocity (W) data, collected by a moored ADCP over more than two and a half years, was aimed to obtain a picture of the migratory behaviour of zooplankton at the daily and the seasonal scale, in relation to the alternation of day and night, to the seasonal stratification of the water column, to dissolved oxygen concentration and blooms of primary production. The

investigated area belongs to an oligotrophic region, which is characterized by a predominant northward current. Seasonal variability of the thermohaline characteristics evidences marked differences between the stratified water column in summer and unstratified water column in winter. Chlorophyll fluorescence gradually increase in the whole upper layer while approaching spring, and the deep chlorophyll maximum (DCM) is undergoing a clear seasonal cycle, being deepest in summer and autumn, and becoming shallower, but more intense, in winter and spring. Along with light and food availability,

stratification and DCM depth are potentially relevant drivers for the seasonal differences of zooplanktonic migratory patterns. The most significant migrations of zooplankton in the Corsica Channel occurs at sunrise (downward) and at sunset (upward). DVM is well recognizable in the intermediate and upper layers and less in the deep one, probably because of the presence of non-migrating epi-benthic or benthopelagic organisms. The night-time biomass increase in shallow layers and its decrease in deep layers is due to nocturnal feeding on phyto- or even zooplankton in the euphotic layer, as done by strong migrators, like

e.g. some chaetognaths (Pearre, 2003). The net samplings evidenced copepods as the most abundant group, followed by other taxa, mainly appendicularians and chaetognaths, and by cladocerans. The zooplankton night-time descent is a well-known behavioural pattern (nocturnal sinking), when sated organisms sink passively due to their higher body density.





At the daily scale, alternation of high and low MVBS is clear in the surface layer. Especially during blooming periods, it peaks from 2 to 4 hours before sunset, a pattern that is consistent with the seasonal presence of twilight migrating organisms. In the intermediate layer MVBS has a more pronounced daily pattern than in the surface layer, with nocturnal high backscatter strength and diurnal MVBS minima. The patterns of descent and ascent are clearly observed throughout the whole period and follow closely the seasonality of sunrise and sunset times. In the deep layer MVBS is very high during the whole experiment with small differences between day and night. Here, daily MVBS values are slightly higher than nocturnal values (the opposite occurs in the intermediate and surface layers), suggesting that during the day there are organisms that migrate from above to high depths (> 350 m).

At the seasonal scale, acoustic backscatter clearly presents periodic oscillations, being higher between late winter and early spring. This bloom in secondary production involves the whole investigated water column and appears to be stronger in deep layers. The bloom is linked to the alternation of stratified and mixing conditions in the water column (MVBS increases when stratification is weaker and the thermocline is almost absent), to the DCM depth, as well as to the seasonality of phytoplankton blooms. In order to identify differences in migratory behaviour according to food availability and light conditions, the blooming and the no-blooming periods have been studied separately. During both periods there are clear upward and downward motions at sunset and sunrise, respectively. However, the blooming period is characterized by a downward movement in the deeper layers and an upward movement in the upper layers throughout the day. During the no-blooming period, biomass maxima split along the water column, with one group of organisms located close to the DCM and the other one in the deep layer (below 300 m). In the course of the day the mid-depth biomass minimum becomes thicker, expanding mainly towards the deeper levels. The superficial group, close to the DCM, is especially evident during the no-blooming period, because of the shallower thermocline and the stronger irradiance during summer (as found also by Pinot and Jansà, 2000). During the blooming period the bimodal distribution of MVBS is weaker and the MVBS minimum layer is thinner and resides at shallower depths if compared to the non-blooming period. In addition, the day-to-night differences in biomass of this layer are less pronounced during the blooming period. Thus, it appears that in the investigated water column DVM is intensified during the blooming period. Consistent with the hypothesis of Hardy and Gunther (1935) and Huggett and Richardson (2000), high food availability results in intensified migration, while scarce food availability results in less intense migration, given the necessity to feed in surface layers also during the day (in spite of the predation risk) in order to compensate for food lack. It is noteworthy, however, that the observed differences between the two periods might not be only correlated to the food availability, but even be a consequence of a community shift or of other seasonally changing environmental factors, e.g. stratification, thermocline depth and position of the DCM.

A spectral analysis applied on both MVBS and W time series confirms the predominance of nocturnal DVM behaviour in this area. Still, other migration patterns (twilight and reverse) could be identified, probably performed by a minority of organisms. The 24 hours peak in the spectrum is linked mainly to nocturnal DVM, while the 12 hours peak is thought to be due to the contemporary presence of organism performing reverse DVM. Other peaks at higher frequencies are linked to different migration patterns along the migratory *continuum* defined by Haney (1988). The spectral analysis performed on the low-passed



time series evidenced peaks at the frequency of the moon cycle (28-30 days), of the four seasons (80-96 days, visible at all levels, but less intense at depth), and of periods in the range 160-193 days (depending on the depth) that might reflect the broader blooming and non-blooming periods.

Bozzano et al. (2014) found that in the shallow water column (0-80 m) of Ligurian Sea zooplankton biomass follows the
primary production signal with a delay of about 1 month in the Ligurian Sea, a result that is consistent with the finding of the present study, with primary production peaks preceding the peaks in shallow secondary production by about 3 weeks in the Corsica Channel. The absence of any temporal lag when comparing deep MVBS vs. Chl *a* in the Corsica Channel is somewhat unexpected, but according to Madin et al. (2001), if the bulk of zooplankton within a water column is composed by vertical migrators, its growth dynamics are not necessarily only coupled to surface primary production. Other studies have shown that
biomass peaks are often coincident (no lag) with chlorophyll maxima (e.g. Jiang et al., 2007).

Knowledge about zooplankton migratory patterns, especially on long time scales (seasonal to interannual), is severely limited because of the difficulties related to net sampling (particularly in the open sea) and to time-consuming taxonomic determinations. Zooplankton plays a pivotal role in the marine food web, biological pump and carbon sequestration, therefore an automatic measurement system with high temporal and spatial coverage, provided by the ADCP, greatly contributes to the
understanding of zooplankton distribution along the water column in different seasons and at different hours of the day, information that are relevant for the modelling of the complex marine biogeochemical mechanisms in which zooplankton is involved. Long time series of acoustic data allows to shed light on scales not resolved by traditional net sampling and this application is a good example of intense exploitation of existing data sets for multiple purposes.

**Acknowledgments**

The authors thank the Captains, the crews and the technicians of the Italian vessels R/V Urania and R/V Minerva Uno. This work was supported by the project OCEAN-CERTAIN (GA# 603773) and JERICO NEXT (GA# 654410) of the European Commission, SSD Pesca project of the Ministry of Economics and Finance; RITMARE flagship project of the Ministry of Education, Universities and Research and the Italian Ministry of Environment for the implementation of the Marine Strategy Framework Directive in Italian waters.

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

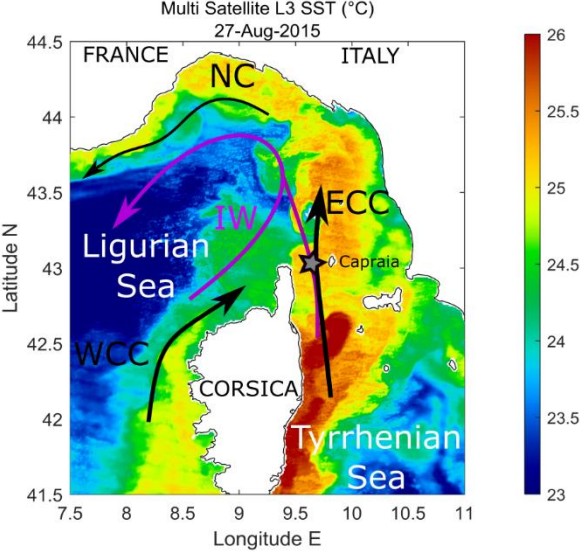

**Figure 1: Map of the area. Main current features (ECC=Eastern Corsica Current; WCC=Western Corsica Current; NC=Northern**
**Current; IW=Intermediate Water) and the position of the moored ADCP (star), are indicated. In the background (colour coded), the SST field from a sample day (27 Aug 2015, °C) is provided to highlight the mesoscale and frontal systems (source CMEMS).**





**Figure 2: (a-d) Vertical profiles of potential temperature, salinity, dissolved oxygen and chlorophyll fluorescence, respectively, from CTD casts carried out during servicing at the mooring location; (e-f) daily vertical profiles of temperature, dissolved oxygen and chlorophyll concentrations, respectively, between 0 and 180 m, from December 2015 to March 2016, as recorded by the moored profiler.**





**Figure 3: (a) Time series (2-hourly) of vertical profiles of MVBS (mean volume of backscatter in dB re(4π*m)-1 ) anomalies (referred to the mean profile of the entire dataset) from April 2014 to November 2016; (b-c) MVBS (in dB re(4π*m)-1 ) and W (mm s-1 ) variations in time as a function of the hour of the day (UTC) at 97m, with the time of sunset and sunrise superimposed (black lines); (d-e) same as (b-c) but at 209 m; (f-g) same as (b-d) but at 353 m.**





**Figure 4: Monthly mean (a) vertical MVBS profiles and their standard deviation (b); (c-d) mean MVBS and W during non-blooming periods (approximately between May and November) as a function of depth and of the hour of the day; (e-f) mean MVBS and W during blooming periods (approximately between December and April) as a function of depth and of the hour of the day. Bluish and reddish arrows in (d) and (f) indicate main downward and upward motions, respectively.**







**Figure 5: (a) Power spectrum of MVBS (high frequency range) at three selected bins (97m, 209m, 353m); (b) same as (a) but for W; (c) power spectrum of the low-passed MVBS time series (low frequency range) at three selected bins (97m, 209m, 353m); (d) same as (b) but for W.**




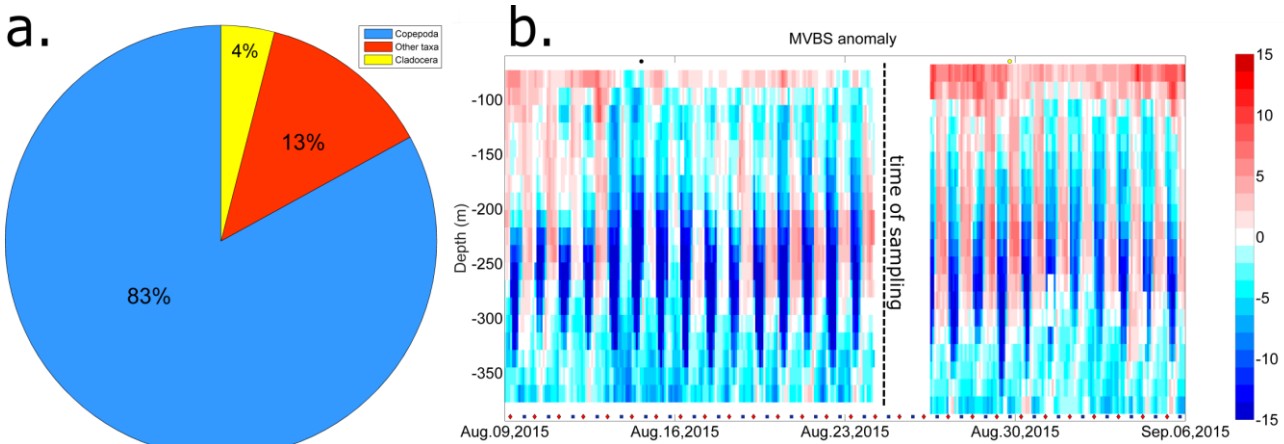

**Figure 6: (a) Mean zooplankton abundances (in %) at the sampling sites in August 2015. (b) MVBS anomalies (in dB re(4π\*m)$^{-1}$)**
**between August 9th and September 6th 2015. Timing of new moon (black dot, above the graph), full moon (yellow dot, above the**
**graph), sunrise (red diamond, below the graph) and sunset (blue square, below the graph) are indicated.**

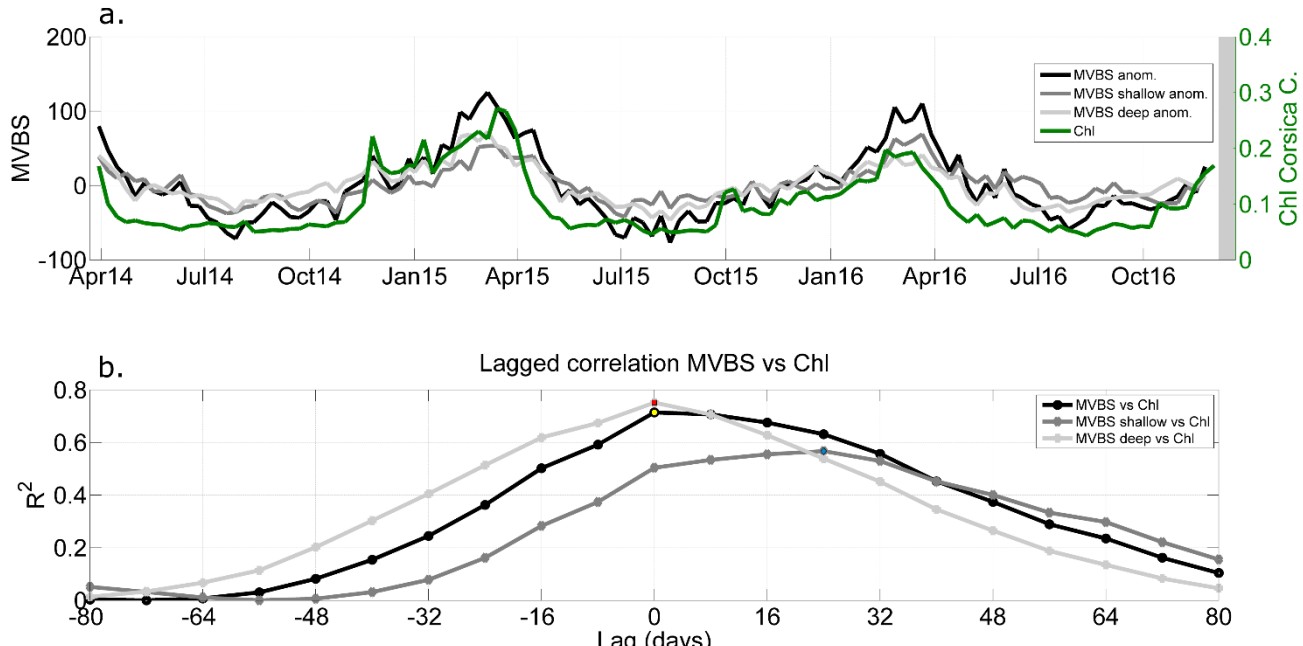

**Figure 7: (a) Time series of integrated anomalies of MVBS (whole water column), MVBS in the layer 73-201m (MVBS shallow),**
**MVBS in the layer 201-378 m (MVBS deep), surface chlorophyll *a* concentration (Chl *a* mg/m3) at the mooring location in the**
**Corsica Channel; (b) lagged correlation analysis between MVBS (whole water column, shallow, deep) and Chl *a*. The yellow circle**
**and the red square indicate the maximum correlation, at lag=0 days, between MVBS (top-bottom) and Chl *a* and MVBS deep and**
**Chl *a*, respectively, while the blue diamond indicates the maximum correlation between MVBS shallow and Chl *a*, at lag=24 days.**





| Deployment | 1 | 2 | 3 | 4 | 5 | 6 | 7 |
|---|---|---|---|---|---|---|---|
| First day | 05/04/14 | 28/11/14 | 21/03/15 | 27/08/15 | 09/12/15 | 21/03/16 | 22/07/16 |
| Last day | 24/11/14 | 19/03/15 | 23/08/15 | 06/12/15 | 19/03/16 | 20/07/16 | 26/11/16 |
| ADCP depth (m) | -395 | -400 | -400 | -411 | -400 | -400 | -400 |
| B (m) | 7.04 | 7.04 | 7.04 | 7.04 | 7.04 | 7.04 | 7.04 |
| Bin number used * | 19 | 19 | 19 | 19 | 19 | 20 | 19 |
| Depth Range (m) | 372-68 | 376-72 | 376-72 | 387-67 | 376-72 | 376-72 | 376-72 |
| Ensembles | 2815 | 1360 | 1878 | 1252 | 1234 | 1470 | 1544 |
| Values discarded | 318 | 480 | 77 | 463 | 780 | 54 | 178 |
| L (m) | 17.16 | 17.16 | 16.97 | 17.42 | 17.04 | 17.04 | 17.04 |
| D (m) | 16 | 16 | 16 | 16 | 16 | 16 | 16 |
| C (dB) | -159.1 | -159.1 | -159.1 | -159.1 | -159.1 | -159.1 | -159.1 |
| $R_0$ (m) | 1.3 | 1.3 | 1.3 | 1.3 | 1.3 | 1.3 | 1.3 |

**Table 1: Deployment characteristics: the depth -400 mis a nominal depth, while -395 m and -411 m is a mean value of the continuous record of the ADCP pressure sensor; the blank is the distance between transducer and the first bin; deployments 1 and 4 had a slowly subsidence, respectively 50 cm in 234 days and 30 cm in 102 days. *Out of 28.**

| Taxon | Group | N $m^{-3}$ sample#1 | N $m^{-3}$ sample #2 | Mean % |
|---|---|---|---|---|
| *Clausocalanus* spp. | COP | 74.22 | 153.31 | 38.96 ± 1.91 |
| *Oithona* spp. | COP | 26.59 | 49.95 | 13.11 ± 0.23 |
| **Appendicularia indet.** | OTH | 8.22 | 12.60 | 3.56 ± 0.59 |
| *Oncaea* spp. | COP | 9.19 | 11.29 | 3.51 ± 1.18 |
| *Paracalanus* spp. | COP | 1.21 | 15.20 | 2.81 ± 2.37 |
| **Chaetognatha** | OTH | 6.77 | 8.69 | 2.65 ± 0.80 |
| *Calocalanus* spp. | COP | 3.14 | 11.29 | 2.47 ± 0.96 |
| *Temora stylifera* | COP | 6.77 | 7.58 | 2.46 ± 1.00 |
| *Ctenocalanus vanus* | COP | 5.56 | 8.69 | 2.44 ± 0.37 |
| *Pleuromamma* spp. | COP | 5.08 | 8.69 | 2.36 ± 0.20 |
| *Corycaeus* spp. | COP | 6.29 | 6.08 | 2.12 ± 1.11 |
| *Nannocalanus minor* | COP | 4.59 | 6.95 | 1.98 ± 0.35 |
| *Pseudoevadne tergestina* | CLA | 0.24 | 10.42 | 1.83 ± 1.83 |
| *Evadne spinifera* | CLA | 0.97 | 9.12 | 1.73 ± 1.34 |

**Table 2: Contribution of the most abundant species/taxa at the two sampling sites in number of individuals (N) per $m^3$ of water (COP: Copepods, CLA: Cladocerans, OTH: other taxa).**