# Peer review of "Zooplankton diel vertical migration in the Corsica Channel (northwestern Mediterranean Sea) detected by a moored ADCP"

_Ocean Science, 2018_

## Referee Comment (RC1) · Anonymous Referee #1 · 14 Nov 2018

General comment: the paper deals with the analysis of backscattered acoustic ADCP data in the Corsica Channel during a period of two and half years to provide understanding on zooplankton behavior and evidence of its vertical migration. The paper contains interesting analysis and findings, but it's lack in some parts and in some discussions. It seems to me that it has been written in a hurry, neglecting some aspects and argumentation on several items with the consequence of not being always clear and correct. Moreover, biological measurements are not really linked to other data. There are several main corrections to do or parts to explain. As the items treated in the paper are interesting, I recommend the publication after all the main following issues have been addresses.

[Figure]

Please also note the supplement to this comment:
https://www.ocean-sci-discuss.net/os-2018-92/os-2018-92-RC1-supplement.pdf
* * *
[Figure]

**Supplement:**

**Title:**
"Zooplankton diel vertical migration in the Corsica Channel (north-western Mediterranean Sea) detected by a moored ADCP"
**by** Davide Guerra, Katrin Schroeder, Mireno Borghini, Elisa Camatti, Marco Pansera, Anna Schroeder, Stefania Sparnocchia, Jacopo Chiggiato.

**General comment:** the paper deals with the analysis of backscattered acoustic ADCP data in the Corsica Channel during a period of two and half years to provide understanding on zooplankton behavior and evidence of its vertical migration. The paper contains interesting analysis and findings, but it's lack in some parts and in some discussions. It seems to me that it has been written in a hurry, neglecting some aspects and argumentation on several items with the consequence of not being always clear and correct. Moreover, biological measurements are not really linked to other data. There are several main corrections to do or parts to explain. As the items treated in the paper are interesting, I recommend the publication after all the main following issues have been addresses.

**Main points:**

1) There is at least one other recent publication on this item in the Mediterranean Sea (page 3, lines 25-27), that is the 2018 paper by Ursella et al. published in Progress in Oceanography on the Southern Adriatic Sea.
2) Potiris et al. 2018, and Pinot et Jansà 2001 also studied the link between DVM and lunar cycle (page 4, line3).
3) At lines 12-18, page3, it is not totally clear what is referred to the whole Mediterranean Sea and what to the Ligurian (also the reference list at line 13 is mixed, but then you speak of the Mediterranean, with a parenthesis on the Ligurian). Please rewrite the sentence.
4) At page 4, Line 10 you write: "to determine how much zooplancton"; this sentence means that you are able through backscattering energy data to measure quantitatively how much zooplancton is present, that is not true, as you also mentioned few lines above. Please change.
5) At line 12, page4, you write: "to identify the drivers": this is a final sentence. As the driving mechanisms of DVM are not totally understood, I would suggest a softer sentence: "to identify the possible drivers". The same at lines 18-21, page 9: the sentence it is very definitive/strong and should be softened and contextualized.
6) you speak of two general and widely accepted assumptions in zooplankton studies (page 5, lines 29-31), but this is not really true. In reality, the sentence found in Heywood 1996 has a slightly different meaning from the one in your text. He says:" For vertical velocities, the water upwelling or downwelling is usually small under general oceanic conditions, except during events such as internal waves... ". I think you should better explain why in your case you can consider the upwelling/downwelling negligible, or change the sentence. Moreover, the second assumption is not generally true: in the case of strong phyto blooming, the layers interested by it, "produce" quite strong signal. The same happens in zones rich in particulate as it could be the layer near the bottom. Anyway, there is no reference for this second assumption. Please explain.
7) the sentence at page 6, Lines 1-2, is not a consequence of the previous one. Moreover, data of zooplankton biomass are not "obtained by the ADCP". Please rewrite the sentence.

8) The paragraph at page 6 from line 25 to line 31 is quite confusing and it should be rewritten. There are not-explained variables in the definition of the two slant range limits, and also the reference is inappropriate. Moreover, the sentence at lines 27-28 is quite twisted. In addition, which "values detected" (line 29) do you mean? As it is, sentence at lines 30-31 is not appropriate as you use eq. 2 to calculate R. Finally, as state by Deines 1999 and Bozzano et al. 2014, the lower limit for the slant range is defined as pi*Ro/4, in order to be used it in the formula of backscatter coefficient (Sv) and not as a general criteria of quality control. Therefore, I would move all this discussion in the following paragraph after definition of R, also for clarity.

9) Do you really have percent good greater than 90% also for data during the day in the parts of the water column where zooplankton has migrated away?

10) The paragraph 3.2 is nested and difficult to follow. Why don't you give formulas 3 and 4 when you mention them the first time at lines 14-15? Eq.2 and 3 are not correct, maybe typing error. Please re-write the paragraph. Moreover, why do you use the formula by Deines 1999 to calculate Sv instead of the upgraded/corrected one suggested by Gostiaux and van Haren 2010 (also in Bozzano et al. 2014)? Please explain.

11) At line 18, page 9, you speak of biomass, but previously, at page 5, you said that you assume that the signal comes only from zooplankton. In many other parts of the text you use biomass in different meanings; this generates some misunderstanding on the word biomass. However, this item should be better explained in the whole text and/or define the terms at the beginning. Please explain and change accordingly.

12) lines from 21 to 28, page 9, are superfluous.

13) Why do you speak of Deep Chlorophyll Maximum if it is seen in the surface layer (line 3 page 10)? The same in conclusions.

14) At line 19, page 10, you say that you use w and Sv "to characterize different migratory behaviours of different zooplanktonic migrator groups", but it doesn't seem to me that you perform this kind analysis, except saying that there are probably two different communities at surface and bottom. Moreover, the following sentence ("To this aim....") is very generic and should be rephrased and the concepts better explained.

15) The lack of information you mention at page 10 line 27 concerns w and MVBS not biomass and migration, except as a consequence. It is quite confusing to a reader.

16) At page 10, line 30, you speak of surface values, but you have just said that there is a lack of data in the surface layer. As this misunderstanding with the term "surface" is found quite often in the text, please fix it throughout the text.

17) at line 31 page 10 you write "since MVBS is a proxy": since this is your assumption and not a general one, it should be changed to "since we use MVBS as a proxy"

18) Why do you affirm that the behaviour observed in MVBS (lines 8-9 page 11, fig.3b, surface layer) is consistent with twilight migrating organism? It is not consistent with the definition you give in the introduction. The same is found at line 18 when speaking of intermediate layer, at line 24 page 13 and in conclusions. Please explain and/or change.

19) It seems to me that in the w plot (fig.3c) the persisting positive values are better seen in January-February than February-March (line 11 page 11).

20) The sentence at line 22-23 of page 11 is not totally true (not for all periods daily vales are slightly higher...). And, are you sure there is no effect of the bottom (like resuspension or particulate) at this quote? please explain.

21) I am not so sure that in Fig.4a the MVBS has a peak in February –March that involves all the water column as you write at line 1 page 12. Maybe in March, but February is not very different from April, except at about 300m.

22) Are you sure that the reference of Pinot et Jansà 2001 at line 10 page 12 is correct? Their measurements reach 220m depth.

23) It would be clearer if you define what you mean by blooming period and no-blooming period, at the beginning of the discussion on the differences in MVBS between the periods (i.e. end of page 11). Moreover, it is not clear what do you mean with the definition of the blooming and no-blooming periods given at page 12 lines 19-22. How do you calculate the periods? Please explain.

24) At line 6 page 13 you mention the fact that the timing of the downward motion in the blooming period is later than in the no-blooming situation due to the later sunrise. But what about the upward motion that happens at the same time in the blooming and no-blooming periods? And what is the timing of sunset in the blooming period: the time written in red in the figures? This is also related to what you write at lines 22-23: it would mean a different timing of reverse migration in the two periods, i.e. 4 hours and 2 hours after upward motion. Please explain.

25) It is hard to understand your affirmation at lines 7-9 page 13, after the discussion just done: DVM is not just presence of more zooplankton in the water column (here again, are you using the term "biomass" instead of zooplankton?); moreover, the upward vertical velocities are stronger in the blooming period, but during the no-blooming period the downward velocities are stronger. What do you exactly mean with "DVM is intensified"? Please explain.

26) The affirmation ("During both periods…") at lines 23-24 page 13, is not so evident to me when looking at figure 4d and 4f: in the no-blooming period it is really weak and should be taken with caution taking into account errors; moreover, these values cover the entire daytime.

27) In order to calculate the FFT, did you interpolate linearly the time series between one deployment and the other? Are you sure that the peaks you find in Fig. 5a and 5b are related to physical phenomena and are not fictitious features? And what about the error bar? Because some of the peaks are really small. The 12-hour peak at 353m is quite evident in w power spectra (lines 32-33 page 13). What do you mean with "taken singularly" at line 33 page 13? Moreover, your discussion at the beginning of page 14 is not convincing me: the reverse migration should be masked by the nocturnal one if it happens exactly at the same time and it is weaker (the bins measure the average movement). Please explain it and give more evidence. Also, the discussion on 4.75 and 8 hours peak (lines 8-10 page 14) is not convincing: the variability you discuss seems not to be a cyclic one with that period. Finally, the spectra of low pass data contain peaks that sometimes are not very evident, and as there are no error bars it is difficult to distinguish them from the surroundings. Please re-do all this part regarding power spectra.

28) I do not understand what is the sense of table2 with the list of all the species, if this feature is not used for the discussion in relation to MVBS and w, and if it is just a snapshot of a summer situation. Also, the small discussion at page 14 is superfluous. The affirmation at lines 30-31 is quite strong and partially not true (evidences of the contrary are found in literature).

29) It is not evident to me the descent during the day between 150 and 250m (fig.6b) as described at line 4 page 15. Moreover, at lines 6-7 page 15, you explain the migration from 100 to 300m citing two references, but you do not say whether you found these organisms in your sampling. You should use your data at least in this discussion. Finally, the sentences at lines 9-12, page15, should be better explained: where do you see the

zooplankton descent? Which behavior do you mean? A reference for it is needed. Here you use "biomass" for phytoplankton.

30) From line 31 page15 to line 13 page 16, you try to explain the unexpected result (zero-lag correlation) with different considerations. It is not clear to me why one of these arguments is the lack of data in the very surface layer if the correlation is the surface layer is ok (by the way, which is the depth of the euphotic layer?). Moreover, why do you cite Warren et al 2004 if they found no correlation or negative one, contrary to your results? Finally, why the changes in the amount of zooplankton should be related to lateral current only in the bottom layer and not in the surface one where the correlation is as you expect? Maybe the MVBS is not always a good proxy for zooplankton biomass, in the sense that it can include other signals? Please, rewrite this part explaining better the concepts.

31) At page16 line you mention an analysis of the behavior of zooplankton in relation with oxygen concentration, but in reality, this kind of analysis is not performed.

32) Taking into consideration all the points above, please change conclusions.

**Minor changes:**

**Abstract:**
Line 14: "Biomass evolution": maybe do you mean "biomass distribution"?
Line 20: cancel "near"
Line22: "others" is quite too general. Please rewrite the sentence.

**Introduction:**
Page1:
Line 26: "At dawn…." It seems a general feature, indeed it is just one type of migration. Please rewrite the paragraph.
Line 31: you are speaking of twilight migration, aren't you? It is not clear.
Line 28: does phytoplankton perform vertical migration?
Page 3:
Line3: instead of "an ADCP" write "an upward-looking ADCP"
Line 15: cancel "are"
Page 4:
Line 5: please correct "by the depth of the depth of"
Line 7: change "calibration" in "calibrate"

**2. Study Area:**
Page 5:
Line 4 and 5: I think that the units are m/s and not cm/s.

**3. Material and Methods:**
Page 5:
Line 20: change "proportional to how fast particles move and it is used to infer the velocity" in "proportional to the velocity of the moving particles and it is used to infer the speed "
Line 23: change "how much sound reflection" in "how much of the sound reflected signal"
Lines 23-26: what you are saying is certainly true, but there is some confusion on the terms you use here (reflection and scatter) and above/below (back-scatter). Please, try to uniform the language.

Page 9:
Line 2: add ":" after "These parameters are"
Lines 2-10: the list of parameter would be more readable if a list number/letter is added (i.e. i) ii) etc.) or it is listed in bullets.
Line 4: cancel "here"
Line 8: cancel "here"
Lines 9-10: moon phases are obtained from where?

**4. Results and Discussion**
Page 10:
Line 13: "Fig 2d-2e" should be "Fig.2e-2g"
Line 24: the acronym has already been defined at page 7.
Line 29: add "approximatively" before "between"
Line 30: change "whole" in "the greatest part of the";
Page 11:
Line 3: "Less evident in fig.3a": I think that it is impossible to see the daily cycle in this panel.
Line 11: change "persisting" in "quite persisting".
Line 19: change "very high" in "quite high"
Line 21-22: cancel "which is below the depth of the ADCP."
Line 24: change "is much lower" in "is hardly seen", also because of what you write few lines below at line 27 ("is not clearly correlated with sunlight etc…").
Line 29: change "from noon to sunset and" in "from noon to sunset in some periods and"
Lines 30-32: the sentence is redundant. Please rewrite it. Moreover, what are DVM parameters?
Line 33: cancel "integrated over the whole investigated water column": the figs.3a and 4a show MVBSs that vary along the water column.
Page 12:
Line 14: cancel "which the ADCP data ….": it is a repetition
Page13:
Line20: change "Fig 3b-3g" in "Fig3b-g"
Line 20: what is "Fig 4-4f"?
Page 14:
Line 9: maybe "Fig.4d"?

**5. Conclusions**
Page16:
Line 33: a reference for the last part of the sentence would be appreciated
Page17:
Line 1: change "surface" with "upper"
Line6: change "daily" with "diurnal"
Line 20: maybe 2000 is 2001?

**Figures:**
Fig. 3: in panels b→g numbers, letters and labels are unreadable. Also, the lines with times of sunset and sunrise are difficult to see.
Fig 4c→f: numbers and letters are too small and units are missing.
Fig. 5: units on the y axis are missing.

Fig. 6 the moon, the sunset and the sunrise symbols are not visible. Fig 6b can be a bit larger and 6a smaller. The use of symbols for sunset and sunrise at the base of the plot makes the plot difficult to interpret.

In general: units are missing in various figures.

**References:**
Potiris et al. 2018: the reference is not complete
Ringelberg 2009: the reference is not complete

---

## Referee Comment (RC2) · Anonymous Referee #2 · 20 Nov 2018

Title

Zooplankton diel vertical migration in the Corsica Channel (north-western Mediterranean Sea) detected by a moored ADCP

Authors

Davide Guerra, Katrin Schroeder, Mireno Borghini, Elisa Camatti, Marco Pansera, Anna Schroeder, Stefania Sparnocchia and Jacopo Chiggiato

General comments

The paper by Guerra et al is a study of diurnal vertical migration of zooplankton in

the Corsica Channel observed with an Acoustic Doppler current profiler for a period of about two and a half years. The study produced interesting results about the vertical and temporal variation of zooplankton distribution and its relation to environmental conditions. The introduction and methodology sections are in general well written. However, some important information regarding the statistical analysis is missing in the methodology section. The results and discussion section requires several changes and a few additions, as some of the text is not easy to follow and understand, and some of the text is not clearly supported by the present graphs. The length of conclusions section should be significantly shortened, as it is largely a repetition of the results and discussion section, through a more synthetic writing. As the results are interesting, I suggest publication after the issues presented below are addressed.

Specific comments

1. Although authors appreciate that the MVBS is only a proxy of zooplankton biomass, they use the term biomass to refer to variations in MVBS. Biomass should be replaced with absolute backscatter or another appropriate term to avoid reader confusion, as details regarding their difference are given only in later sections.

2. Results should be presented in the past tense.

3. Please add units that are missing in several figures and use equation editor for the units in figure captions, not text.

4. Please consider adding density profiles to figure 2 and refer to the pycnocline instead of the thermocline when mentioning stratification.

5. Please distinguish between primary (phytoplankton) and secondary (zooplankton) bloom throughout the text (or at least once in each paragraph). In some cases, it was obvious from context which one was meant, in others, it was a bit confusing.

6. p.3, l.21-24: Please consider expanding a bit the discussion on the drivers of DVM.

7. p.3, l.25: van Haren, J. Plankton Res., 2014 and Ursella et al, Deep-Sea Res., 2018

are two additional ADCP studies on zooplankton in the Mediterranean Sea. Please consider including them.

8. p.3, l.28: "...to infer the composition in the Ligurian Sea...". This is incorrect, they only suggest that a change in composition is probable. Please remove.

9. p.4, l.6-8: Please consider including Brierley et al, Deep-Sea Res., 1998.

10. p.5, l.18-24: Please consider moving "The operating ... (Thomson and Emery, 2014)." to introduction and merge the rest of this paragraph with the next one.

11. p.5, l.32: Do you mean composition instead of "...consistency..."?

12. p.6, l.1-2: "Therefore, ... quantitative." Repetition (also on p.4, l.6). Please consider removing.

13. p.6, l.5: Please consider moving "The four ... signals." to the previous paragraph which explain the operation principles.

14. p.6, l.6: Please replace "...is upward looking..." with "...is placed at an upward looking position...". The way is stated, one might understand that this particular ADCP can be used only in an upward looking position, which is incorrect.

15. p.6, l.16-20: This paragraph could be removed.

16. p.6, l.27: Please explain symbols H and $\theta$.

17. p.7, l.1: There is one PG per transducer and an average PG. Which one was used? The average, the minimum of all separate transducers or something else?

18. p.7, l.13: What data were used for the calculation of the absorption coefficient $\alpha$?

19. p.8, l.18: Perhaps you meant "...complemented..." instead of "...integrated..."?

20. p.8, l.22: Please take also into consideration that large organisms can escape the 200 $\mu$m mesh. Moriarty et al, Earth Syst. Sci. Data, 2013.

21. p.9, l.9: The sentence seems incomplete.

22. p.9, l.12: Please provide more information on spectral analysis. Are the data detrended, windowed, block-band averaged, which is the number of segments for the spectral estimate, what is the segment overlap? These are necessary for the calculation of confidence intervals.

23. p.9, l.15: Please consider replacing "...verify..." with "...investigate...".

24. p.9, l.18-28: "". Repetition (also on p.4, l.15-24). Please consider removing.

25. p.10, l.4: Please consider moving the DCM definition to p.5, l.10-15 and add some information regarding its variability from literature.

26. p.10, l.10: It is not clear to me which this interface is.

27. p10, l.18-23: I think that "Vertical ... range." should be moved to methodology.

28. p.10, l.26: "All considerations ... bottom." Repetition (p.7, l.3-4). Please consider removing.

29. p.11, l.3: The daily cycle is embedded in the plot, but it is not distinguishable. Please consider including a representative subplot with time span of a few days.

30. p11, l.7: "June-July 2016". I think it's around April, not June-July.

31. p11, l.9: "...a pattern ... organisms." Please add reference.

32. p.11, l.22: "...daily values are slightly higher than nocturnal values...". Please include a supporting graph or mean daily and night MVBS values.

33. p.11, l.33-34: Please include a plot of integrated MVBS as the argument is not evident from figures 3 and 4.

34. p.12, l.3: I think it's "...intra-annual..." instead of "...interannual...".

35. p.13, l.5 and figure 4c-4f: Since light intensity was found to be the governing factor

controlling DVM (e.g. figure 3), the x axis should be hours relative to sunrise and sunset instead of hour of the day for the W ADCP to be more representative of actual zooplankton migrating velocity. Qualitatively, the results will be the same as those in figure 4d and 4f, but I expect that the duration of upwards and downwards motion will last less time than is shown in the present plot. Please consider, either including a plot with such an x axis, or adding some text explaining that the vertical velocity values are not optimally presented in this plot.

36. p.14, l.4: "...or does not involve a large number of organisms." This applies only (and partially) to MVBS spectra. The way it is stated both MVBS and W are meant, which is incorrect.

37. p.14, l.6 and 8: The smallest annotated period in figure 5 is 4.45 hours. It is 4.75 hours in the text.

38. p.14, l.30-31: "The community is essentially composed by organisms that do not migrate significantly...". Please add reference.

39. p.15, l.10: Please replace "...which is accompanied..." with "...which is possibly accompanied..." as the lack of surface data hampers further investigation.

40. p.15, l.11-12: Please add reference.

41. p.15, l.17: Is the distinguish between shallow and deep layers based on the photic layer depth or on another criterion?

42. p.15, l.27: "...that well correlate...". The Chl-$\alpha$ and MVBS time series should be pre-whitened (i.e. remove autocorrelation) before a conclusion is drawn regarding their degree of correlation.

43. p.15, l.32: Please consider replacing "...a surface value..." with "...an exponentially weighted near-surface value...".

44. p.15-16, l.29-13: Please consider placing the part of literature that supports/contradicts the findings of this study at the beginning of the paragraph and then present possible explanations for this agreement/disagreement. I was confused.

45. Conclusions section: I think that this part should be rewritten to avoid repetition of results. Instead, the relation of the results of the present study with the relevant literature should be stressed. Also, the length of this section should be substantially shortened.

46. figure 1: Please include information about the data set of SST field in the data availability section.

47. figure 4a and 4b: The x axis is month or climatological month? It was not clear to me from the text. If climatological please add this to the axis label. Otherwise, state which year the plot refers to.

48. figure 5: Please add confidence intervals. This is particularly important for the low-passed series (5c and 5d) and subsequent interpretation of results.

49. figure 6a: It seems redundant to me as the three numbers in this plot are already present in the text.

50. figure 7b: Please add confidence intervals.

51. table 1: Please explain symbols in table caption or replace B, L, D, C with blank distance, etc.

52. table 2: I think that this table is redundant, as the useful information of taxonomic analysis has been already presented in the text. Please consider removing.

Technical corrections

1. Please add the data availability section that is missing (required by journal).

2. Bibliography is not formatted according to journal standards. Number of volume and pages are missing. Also, doi representation is not consistent among references (some

are doi:... others are https://doi...).

3. p.3, l.16: Please merge the two sentences or rephrase.

4. p.4, l.5: "...by the depth of the depth...". Typo.

5. p.4, l.7: "...attempts to calibration...". Typo.

6. p.4, l.17: "...Data collections...". Typo.

7. p.4, l.22: Perhaps replace "...is completed by..." with "...concludes with..."?

8. p.5, l.18: Please replace "...as sediments..." with "...such as sediments...". Typo.

9. p.6, l.5: "...increments to each other...". Typo.

10. p.6, l.28: #7 deployment is missing. Typo.

11. p.7, l.3: Perhaps replace "...will be done..." with "...will be made..."?

12. p.9, l.15: Please consider replacing "...results to be..." with "...is a relevant...".

13. p.9, l.31: Please replace "...servicing..." with "...mooring maintenance...".

14. p.9-10, l.31-3: please consider merging the two sentences.

15. p.10, l.1: "...are representative of..." instead of "...represents...". Typo.

16. p.10, l.4: "...in correspondence with..." instead of "...in correspondence of...". Typo.

17. p.11, l.24: Perhaps replace "...much lower." with "...much weaker."?

18. p.14, l.4: "...take place..." instead of "...take places...". Typo.

19. p.14, l.24: Please consider replacing "...by far the most abundant group were the copepods..." with "...the copepods were by far the most abundant...".

20. p.14, l.29: Please consider removing "more" in "...more western...".
[Figure]

21. p.15, l.3: Please consider replacing "...more superficial..." with "shallower".

22. p.15, l.9: "Fig." instead of "fig.". Typo.

23. p.15, l.26: "...are shown..." instead of "...is shown...". Typo.

24. figure 1: Please change "IW=Intermediate Water" to "IW=Intermediate Water pathway" or something similar.

25. table 1. "...-400 mis...". Typo.

---

## Author Comment (AC1) · 16 Jan 2019

Answers (*A*) to reviewer's comments (*R*) are written in *italics*.

**General comments**
The paper deals with the analysis of backscattered acoustic ADCP data in the Corsica Channel during a period of two and half years to provide understanding on zooplankton behavior and evidence of its vertical migration. The paper contains interesting analysis and findings, but it's lack in some parts and in some discussions. It seems to me that it has been written in a hurry, neglecting some aspects and argumentation on several items with the consequence of not being always clear and correct. Moreover, biological measurements are not really linked to other data. There are several main corrections to do or parts to explain. As the items treated in the paper are interesting, I recommend the publication after all the main following issues have been addresses.

**Main points:**
R: There is at least one other recent publication on this item in the Mediterranean Sea (page 3, lines 25---27), that is the 2018 paper by Ursella et al. published in Progress in Oceanography on the Southern Adriatic Sea.
*A: Thank you, the reference has been added and briefly described. It was available online just a few days before our submission, thus we missed it*

R: Potiris et al. 2018, and Pinot et Jansà 2001 also studied the link between DVM and lunar cycle (page 4, line3).
*A: Added*

R: At lines 12---18, page3, it is not totally clear what is referred to the whole Mediterranean Sea and what to the Ligurian (also the reference list at line 13 is mixed, but then you speak of the Mediterranean, with a parenthesis on the Ligurian). Please rewrite the sentence.
*A: we rewrote the sentence*

R: At page 4, Line 10 you write: "to determine how much zooplankton"; this sentence means that you are able through backscattering energy data to measure quantitatively how much zooplankton is present, that is not true, as you also mentioned few lines above. Please change.
*A: We rephrased the sentence, which is now: "allow to know relative abundances of zooplankton present at a certain depth"*

R: At line 12, page4, you write: "to identify the drivers": this is a final sentence. As the driving mechanisms of DVM are not totally understood, I would suggest a softer sentence: "to identify the possible drivers". The same at lines 18---21, page 9: the sentence it is very definitive/strong and should be softened and contextualized.
*A: Done, we accepted the suggestion to use a softer statement. At p. 9 we deleted ''understand'' and put ''improve knowledge about what might possibly drive''.*

R: you speak of two general and widely accepted assumptions in zooplankton studies (page 5, lines 29---31), but this is not really true. In reality, the sentence found in Heywood 1996 has a slightly different meaning from the one in your text. He says:" For vertical velocities, the water upwelling or downwelling is usually small under general oceanic conditions, except during events such as internal waves... ". I think you should better explain why in your case you can consider the upwelling/downwelling negligible or change the sentence. Moreover, the second assumption is not generally true: in the case of strong phyto blooming, the layers interested by it, "produce" quite strong signal. The same happens in zones rich in particulate as it could be the layer near the bottom. Anyway, there is no reference for this second assumption. Please explain.

*A: We have rephrased and added a reference study from which it results that the Corsica Channel does not belong to the group of upwelling/downwelling areas of the Mediterranean. The second is especially an assumption that we made for our work, but that is often done in previous studies, among which we decided to mention one. We also added a small remark on the fact that sound backscatter has also other causes, which we are not able to discriminate. This is the sense of an "assumption"*

R: the sentence at page 6, Lines 1---2, is not a consequence of the previous one. Moreover, data of zooplankton biomass are not "obtained by the ADCP". Please rewrite the sentence.
*A: The sentence is still valid, we have cancelled the term "therefore" and made some other changes. The sentence is now: "In general, information on zooplankton biomass and vertical motion inferred from ADCP data are more qualitative than quantitative".*

R: The paragraph at page 6 from line 25 to line 31 is quite confusing and it should be rewritten. There are not---explained variables in the definition of the two slant range limits, and also the reference is inappropriate.
 Moreover, the sentence at lines 27---28 is quite twisted. In addition, which "values detected" (line 29) do you mean? As it is, sentence at lines 30---31 is not appropriate as you use eq. 2 to calculate R. Finally, as state by Deines 1999 and Bozzano et al. 2014, the lower limit for the slant range is defined as pi*Ro/4, in order to be used it in the formula of backscatter coefficient (Sv) and not as a general criteria of quality control. Therefore, I would move all this discussion in the following paragraph after definition of R, also for clarity.
*A: We have rewritten this part and added the missing definition of two variables. We think that the explanation is now much clearer. We do not understand what is inappropriate about the reference, it is a technical manual for the instrument written by the manufacturer. The order in which equations are presented has also been changed.*
*Concerning the comment about the lower limit of the slant range, the reviewer is right, but in our case no data were rejected because of this, but only because some were exceeding the maximum threshold, which is indeed a general criteria of quality control. Therefore, for the sake of clarity, we split the text, leaving discussion about Rmax here and moving Rmin to the following paragraph for the calculation of Sv.*

R: Do you really have percent good greater than 90% also for data during the day in the parts of the water column where zooplankton has migrated away?
*A: All data where PG<90% were discarded, before analyzing them, so the answer is yes.*

R: The paragraph 3.2 is nested and difficult to follow. Why don't you give formulas 3 and 4 when you mention them the first time at lines 14---15? Eq.2 and 3 are not correct, maybe typing error. Please re---write the paragraph. Moreover, why do you use the formula by Deines 1999 to calculate Sv instead of the upgraded/corrected one suggested by Gostiaux and van Haren 2010 (also in Bozzano et al. 2014)? Please explain.
*A: Following a previous comment, we reorganized the whole part and moved formulas to the right place. Typos were corrected.*
*We did a choice of one of the methods available in literature, also more recent papers than Bozzano et al. still continue to use Deines approach. Also Potiris et al 2018 used Deines formula, and they had an experiment setup more similar to ours, than Bozzano et al. Our aim was not to compare methodologies. In addition, our dataset does not have low signal to noise ratios (<10), for which the method has been developed by Gostiaux and van Haren.*

R: At line 18, page 9, you speak of biomass, but previously, at page 5, you said that you assume that the signal comes only from zooplankton. In many other parts of the text you use biomass in different meanings; this generates some misunderstanding on the word biomass. However, this item should be better explained in the whole text and/or define the terms at the beginning. Please explain and change accordingly.
*A: Thank you for outlining this. We have added zooplanktonic to all "biomass" words where it was necessary.*

R: lines from 21 to 28, page 9, are superfluous.
*A: We cancelled them*

R: Why do you speak of Deep Chlorophyll Maximum if it is seen in the surface layer (line 3 page 10)? The same in conclusions.
*A: DCM is a widely accepted definition which refers to the region below the surface of water with the maximum concentration of chlorophyll. In the study area it is located between 20 and 100 meters, depending on the season. Sometimes it is also called subsurface Chlorophyll maximum, but more frequently DCM, regardless of depth. For clarity we moved this definition to section 3.3.*

R: At line 19, page 10, you say that you use w and Sv "to characterize different migratory behaviors of different zooplanktonic migrator groups", but it doesn't seem to me that you perform this kind analysis, except saying that there are probably two different communities at surface and bottom. Moreover, the following sentence ("To this aim….") is very generic and should be rephrased and the concepts better explained.
*A: Ok, we have canceled the part "to characterize…groups". The sentence "To this aim…" in our opinion explains in a straightforward way that without a proper calibration, we can use MVBS only as an indirect and qualitative indicator of zoopl. biomass. We have therefore left it, rephrasing it a bit: "Without the necessary net samples that would allow a proper calibration, MVBS is considered as an indirect and qualitative proxy of zooplanktonic biomass".*

R: The lack of information you mention at page 10 line 27 concerns w and MVBS not biomass and migration, except as a consequence. It is quite confusing to a reader.
*A: Ok, we replaced biomass and migration with MVBS and W. The consequence is obviously that nothing can be said about zooplanktonic biomass and migration in this layer.*

R: At page 10, line 30, you speak of surface values, but you have just said that there is a lack of data in the surface layer. As this misunderstanding with the term "surface" is found quite often in the text, please fix it throughout the text.
*A: OK, thanks, we replaced ''surface'' with ''in the upper part of investigated water column'' or "upper layer". Checked out also throughout the paper.*

R: at line 31 page 10 you write "since MVBS is a proxy": since this is your assumption and not a general one, it should be changed to "since we use MVBS as a proxy"
*A: Done*

R: Why do you affirm that the behavior observed in MVBS (lines 8---9 page 11, fig.3b, surface layer) is consistent with twilight migrating organism? It is not consistent with the definition you give in the introduction.
The same is found at line 18 when speaking of intermediate layer, at line 24 page 13 and in conclusions. Please explain and/or change.
*A: We removed the "twilight" part from the surface layer discussion. However, what we observe in the intermediate layer is consistent with the definition of twilight migration we gave in the introduction (upward motion right after sunrise can be due to twilight migrators and reverse migrators). In addition, we have cancelled the part at page 13 "During both periods, an upward motion is evident after sunrise, a feature that is characteristic of twilight and reverse migration." Conclusions has been rewritten accordingly*

R: It seems to me that in the w plot (fig.3c) the persisting positive values are better seen in January---February than February---March (line 11 page 11).
*A: There was a problem with the labels in Fig. 3b-f, they were misplaced, the correct ones were those of Fig. 3g. So February-March was correct.*

R: The sentence at line 22---23 of page 11 is not totally true (not for all periods daily vales are slightly higher…). And, are you sure there is no effect of the bottom (like resuspension or particulate) at this quote? please explain.

*A: Actually, we analyzed several (this word has been added) profiles from transmissometers routinely mounted on our CTD-rosette system, and turbidity values at the depths above the ADCP were always very low. We corrected the sentence by adding "except during the zooplanktonic blooming period".*

R: I am not so sure that in Fig.4a the MVBS has a peak in February –March that involves all the water column as you write at line 1 page 12. Maybe in March, but February is not very different from April, except at about 300m.

*A: OK, we wrote only "March"*

R: Are you sure that the reference of Pinot et Jansà 2001 at line 10 page 12 is correct? Their measurements reach 220m depth.

*A: Yes, because besides the difference of depth, we as well hypothesize the possible presence of two communities (like Pinot and Jansà)*

R: It would be clearer if you define what you mean by blooming period and no---blooming period, at the beginning of the discussion on the differences in MVBS between the periods (i.e. end of page 11). Moreover, it is not clear what do you mean with the definition of the blooming and no---blooming periods given at page 12 lines 19---22. How do you calculate the periods? Please explain.

*A: The definition of the two periods has been moved to the part that was at the end of page 11 as you suggested.*

R: At line 6 page 13 you mention the fact that the timing of the downward motion in the blooming period is later than in the no---blooming situation due to the later sunrise. But what about the upward motion that happens at the same time in the blooming and no--- blooming periods? And what is the timing of sunset in the blooming period: the time written in red in the figures? This is also related to what you write at lines 22---23: it would mean a different timing of reverse migration in the two periods, i.e. 4 hours and 2 hours after upward motion. Please explain.

*A: We commented the upward motion already, it happens at the same time but is more intense during the blooming. Sunset and sunrise times varies during the blooming period and the non-blooming period since it is a period of about 6 months. In red is only written the timing that was found to be the most evident in W. We added this information on the figure caption. We can not be more precise because of the 2h sampling, and also because the figures are averages of long periods.*

R: It is hard to understand your affirmation at lines 7---9 page 13, after the discussion just done: DVM is not just presence of more zooplankton in the water column (here again, are you using the term "biomass" instead of zooplankton?); moreover, the upward vertical velocities are stronger in the blooming period, but during the no---blooming period the downward velocities are stronger. What do you exactly mean with "DVM is intensified"? Please explain.

*A: We have added "zooplanktonic" before "biomass" to be clearer. DVM is not the presence of more zooplankton and the comment in fact refers to intensified W values. We replaced the concept of intensified DVM with intensified active upward motion.*

R: The affirmation ("During both periods…") at lines 23---24 page 13, is not so evident to me when looking at figure 4d and 4f: in the no---blooming period it is really weak and should be taken with caution taking into account errors; moreover, these values cover the entire daytime.

*A: You are right, we have deleted the sentence*

R: In order to calculate the FFT, did you interpolate linearly the time series between one deployment and the other? Are you sure that the peaks you find in Fig. 5a and 5b are related to physical phenomena and are not fictitious features? And what about the error bar? Because some of the peaks are really small. The 12---hour peak at 353m is quite evident in w power spectra (lines 32---33 page 13).

What do you mean with "taken singularly" at line 33 page 13?

Moreover, your discussion at the beginning of page 14 is not convincing me: the reverse migration should be masked by the nocturnal one if it happens exactly at the same time and it is weaker (the bins measure the average movement). Please explain it and give more evidence.

Also, the discussion on 4.75 and 8 hours peak (lines 8---10 page 14) is not convincing: the variability you discuss seems not to be a cyclic one with that period. Finally, the spectra of low pass data contain peaks that sometimes are not very evident, and as there are no error bars it is difficult to distinguish them from the surroundings. Please re---do all this part regarding power spectra.

*A: We interpolated with the matlab function inpaint_nans which is based on a PDE that is assumed to apply in the domain of the artifact to be interpolated. Then the PDE is approximated using finite difference methods and is solved for the NaN elements in the array. This method did not produce any peaks in Fig. 5a and b. We added this information in section 3.5.*

*We specified at which periods and depths peaks are really small (added this information where it was missing).*

*We removed "taken singularly", the sentence should be clearer. The meaning was that if reverse migration occurs alone the peak would be at 24 h, the same is true for nocturnal migration. But if they occur both the resulting peaks are at both 24h and 12h. It is quite difficult to explain the 12h peak existence, but the most plausible explanation is that reverse and nocturnal occur both, even though not at the exact same time (which would be rather strange indeed). The other peaks are representative of some cyclic variability, although it is rather difficult to identify which kind of migration is responsible for it. We smooth out the text a bit to take into account that we don't know how much we can trust these peaks. Finally, the spectrum was computed with a straightforward FFT, without segmentation and overlapping. Especially when looking for the long periods, segmentation would not have allowed to detect them. This is why we could not compute the confidence intervals here. Also here we smooth out the text a bit to take into account that we don't know how much we can trust these peaks.*

R: I do not understand what is the sense of table2 with the list of all the species, if this feature is not used for the discussion in relation to MVBS and w, and if it is just a snapshot of a summer situation. Also, the small discussion at page 14 is superfluous. The affirmation at lines 30---31 is quite strong and partially not true (evidences of the contrary are found in literature).

*A: We decided to keep this table, to summarize the net sample findings, which not entirely are described in the text, even if they are just a snapshot, compare to the acoustic data these represent a sort of ground truth, which is important to account for (this remark has been added to the text). In addition, it could be a useful reference for future studies of the communities in this area. It shows that the characteristics of this community are mainly epipelagic, which are not strong vertical migrators, as is evident also from the acoustic data of august. It is not clear to us what the reviewer wants to criticize about L30-31, and to which references in the literature he is referring to.*

R: It is not evident to me the descent during the day between 150 and 250m (fig.6b) as described at line 4 page 15. Moreover, at lines 6---7 page 15, you explain the migration from 100 to 300m citing two references, but you do not say whether you found these organisms in your sampling. You should use your data at least in this discussion Finally, the sentences at lines 9---12, page15, should be better explained: where do you see the zooplankton descent? Which behavior do you mean? A reference for it is needed. Here you use "biomass" for phytoplankton.

*A: We have removed Fig. 6a and now the plot 6b should be more readable. We replaced with "low MVBS levels" instead of "descent". We have specified that the groups of organisms we hypothesized have although not been sampled, stressing that the sample, even if it is necessary ground-truth base, is just a snapshot of a summer situation during day.*

*Line 9-12 page 15: there are probably zooplanktonic organisms in the upper layer we do not see except during their descent when DCM deepens. We add the reference to Fig. 4a where this is visible. We added ''phytoplankton'' before "biomass ''.*

R: From line 31 page15 to line 13 page 16, you try to explain the unexpected result (zero--- lag correlation) with different considerations. It is not clear to me why one of these arguments is the lack of data in the

very surface layer if the correlation is the surface layer is ok (by the way, which is the depth of the euphotic layer?). Moreover, why do you cite Warren et al 2004 if they found no correlation or negative one, contrary to your results? Finally, why the changes in the amount of zooplankton should be related to lateral current only in the bottom layer and not in the surface one where the correlation is as you expect? Maybe the MVBS is not always a good proxy for zooplankton biomass, in the sense that it can include other signals? Please, rewrite this part explaining better the concepts.

*A: This part has been completely rearranged (some results change after correction on the computing method suggested by the reviewer#2). We think that now the part is clearer. We mentioned the fact that satellite data are just surface data because as we know from Fig. 2d the DCM can be as deep as 100 m and, in this case, the phytoplanktonic bloom might not be correctly sampled by satellites or its timing might be different to what is seen from satellites. The depth of the euphotic layer is approx. the lower limit of the DCM, Fig. 2d. Reference to Warren 2004 has been removed as well as the sentence containing it. We do however do not follow the reviewer's reasoning when he says that lateral currents influence only the bottom layer and that MVBS is representing also other signals. Lateral currents are able to influence the whole water column (and this is the sense of what is written in the ms). Nepheloid layers have already been excluded in the paper, based on transmissometer data analysis.*

R: At page16 line you mention an analysis of the behavior of zooplankton in relation with oxygen concentration, but in reality, this kind of analysis is not performed.
*A: Oxygenation has been described for different season, so yes an in depth analysis has not been performed, but it was part of the description of the water mass properties. This part on oxygen has been deleted from the conclusions.*

R: Taking into consideration all the points above, please change conclusions.
*A: Conclusions have been rewritten accordingly*

**Minor changes:**

| Comments | Corrections |
|---|---|
| Line 14: "Biomass evolution": maybe do you mean "biomass distribution"? | *We replace "evolution" with "temporal distribution".* |
| Line 20: cancel "near" | *Done.* |
| Line22: "others" is quite too general. Please rewrite the sentence. | *Replaced with "other factors, like lunar cycle and primary production, are taken in consideration"* |
| Page1: Line 26: "At dawn…." It seems a general feature, indeed it is just one type of migration. Please rewrite the paragraph. | *Replaced with "During nocturnal migration at dawn…".* |
| Line 31: you are speaking of twilight migration, aren't you? It is not clear. | *Replaced with "The typical descent of twilight migration that…".* |
| **Page 2** Line 28: does phytoplankton perform vertical migration? | *Some of them do it, especially dinoflagellate. Phytoplankton plays a role in the vertical flows of matter / energy and the range of autotrophic organisms capable of prolonged direct vertical movements also involves flagellated phytoplankton* |
| **Page 3:** Line3: instead of "an ADCP" write "an upward---looking ADCP" | *Done.* |
| Line 15: cancel "are" | *Done, this part has been rewritten according to your comment #3* |
| **Page 4:** Line 5: please correct "by the depth of the depth of" | *Done* |
| Line 7: change "calibration" in "calibrate" | *Done.* |
| **Page 5:** Line 4 and 5: I think that the units are m/s and not cm/s. | *Yes you're right, thanks* |

| | |
|---|---|
| **Page 5:** Line 20: change "proportional to how fast particles move and it is used to infer the velocity" in "proportional to the velocity of the moving particles and it is used to infer the speed " | *Done.* |
| Line 23: change "how much sound reflection" in "how much of the sound reflected signal" | *Done.* |
| Lines 23---26: what you are saying is certainly true, but there is some confusion on the terms you use here (reflection and scatter) and above/below (back---scatter). Please, try to uniform the language. | *Back-scatter vs backscatter have been uniformed throughout the paper. The confusion between reflection and scattering is not evident to us, since they are two different physical processes. However, it is true that what is commonly called "backscatter" in the water column includes also those particles that reflect the sound wave, and not only those that scatter it.* |
| **Page 9:** Line 2: add ":" after "These parameters are" | *Done.* |
| Lines 2---10: the list of parameters would be more readable if a list number/letter is added (i.e. i) ii) etc.) or it is listed in bullets. | *We decided not to use a list, but keep a text.* |
| Line 4: cancel "here" | *Done.* |
| Line 8: cancel "here" | *Done.* |
| Lines 9---10: moon phases are obtained from where? | *We have added the source "(retrieved from https://aa.usno.navy.mil/data/docs/MoonPhase.php)"* |
| **Page 10:** Line 13: "Fig 2d---2e" should be "Fig.2e---2g" | *Done.* |
| Line 24: the acronym has already been defined at page 7. | *Deleted.* |
| Line 29: add "approximatively" before "between" | *Done.* |
| Line 30: change "whole" in "the greatest part of the"; | *We replaced it with "most of the"* |
| Page 11: Line 3: "Less evident in fig.3a": I think that it is impossible to see the daily cycle in this panel. | *You're right and we removed this, changing this and the following sentence.* |
| Line 11: change "persisting" in "quite persisting". | *Done.* |
| Line 19: change "very high" in "quite high" | *Done.* |
| Line 21---22: cancel "which is below the depth of the ADCP." | *Done* |
| Line 24: change "is much lower" in "is hardly seen", also because of what you write few lines below at line 27 ("is not clearly correlated with sunlight etc…"). | *Done.* |
| Line 29: change "from noon to sunset and" in "from noon to sunset in some periods and" | *Done.* |
| Lines 30---32: the sentence is redundant. Please rewrite it. Moreover, what are DVM parameters? | *We rewrote the sentence.* |
| Line 33: cancel "integrated over the whole investigated water column": the figs.3a and 4a show MVBSs that vary along the water column. | *Done, thanks* |
| **Page 12:** Line 14: cancel "which the ADCP data ….": it is a repetition | *Done.* |
| **Page13:** Line20: change "Fig 3b---3g" in "Fig3b---g" | *Corrected to ''in Fig.3b-g and Fig4c-f''.* |
| Line 20: what is "Fig 4---4f"? | *corrected* |

| | |
|---|---|
| **Page 14:** Line 9: maybe "Fig.4d"? | *Yes. Done, thanks* |
| **Page16:** Line 33: a reference for the last part of the sentence would be appreciated | *We modified the text a bit and added the reference Tarling et al., 2002* |
| **Page17:** Line 1: change "surface" with "upper" | *Done.* |
| Line6: change "daily" with "diurnal" | *Done.* |
| Line 20: maybe 2000 is 2001? | *Done.* |
| **Figures:** | |
| Fig. 3: in panels b→g numbers, letters and labels are unreadable. Also, the lines with times of sunset and sunrise are difficult to see. | *Corrected* |
| Fig 4c→f: numbers and letters are too small and units are missing. | *Corrected* |
| Fig. 5: units on the y axis are missing. | *Corrected* |
| Fig. 6 the moon, the sunset and the sunrise symbols are not visible. Fig 6b can be a bit larger and 6a smaller. The use of symbols for sunset and sunrise at the base of the plot makes the plot difficult to interpret. | *We have modified the sizes of the two plots, and used bigger symbols for the moon phase. We could not make the symbols of sunset and sunrise larger, because of lacking space. If one zooms into the pdf of the ms they result visible.* |
| In general: units are missing in various figures. | *Corrected* |
| Potiris et al. 2018: the reference is not complete | *Corrected* |
| Ringelberg 2009: the reference is not complete | *Corrected* |

---

## Author Comment (AC2) · 16 Jan 2019

Answers (*A*) to reviewer's comments (*R*) are written in *italics*.

**General comments**
The paper by Guerra et al is a study of diurnal vertical migration of zooplankton in the Corsica Channel observed with an Acoustic Doppler current profiler for a period of about two and a half years. The study produced interesting results about the vertical and temporal variation of zooplankton distribution and its relation to environmental conditions. The introduction and methodology sections are in general well written. However, some important information regarding the statistical analysis is missing in the methodology section *(A: These parts have been added when required)*. The results and discussion section requires several changes and a few additions, as some of the text is not easy to follow and understand, and some of the text is not clearly supported by the present graphs *(A: changes have been made according to specific comments below)*. The length of conclusions section should be significantly shortened *(A: has been shortened)*, as it is largely a repetition of the results and discussion section, through a more synthetic writing. As the results are interesting *(A: thank you)*, I suggest publication after the issues presented below are addressed.

**Specific comments**
R: Although authors appreciate that the MVBS is only a proxy of zooplankton biomass, they use the term biomass to refer to variations in MVBS. Biomass should be replaced with absolute backscatter or another appropriate term to avoid reader confusion, as details regarding their difference are given only in later sections.
*A: We have added a sentence where we say that in the following parts of the paper the term zooplanktonic biomass will be used when referring to results coming from MVBS data*

R: Results should be presented in the past tense.
*A: We decided to keep our style, which is used consistently throughout the paper.*

R: Please add units that are missing in several figures and use equation editor for the units in figure captions, not text.
*A: Added, but did not see the need to use equation editor. Some apexes in the text of the captions have been corrected.*

R: Please consider adding density profiles to figure 2 and refer to the pycnocline instead of the thermocline when mentioning stratification.
*A: Since T is leading density, the thermocline is equivalent to the pycnocline. Adding density would therefore not add any useful information and would just take up space. We added a small text in the caption explaining this*

R: Please distinguish between primary (phytoplankton) and secondary (zooplankton) bloom throughout the text (or at least once in each paragraph). In some cases, it was obvious from context which one was meant, in others, it was a bit confusing.
*A: Done, thanks*

R: p.3, l.21-24: Please consider expanding a bit the discussion on the drivers of DVM.
*A: We think that the Introduction is already very long to go further in detail. However, all the relevant previous works are there, so the reader can find the sources.*

R: p.3, l.25: van Haren, J. Plankton Res., 2014 and Ursella et al, Deep-Sea Res., 2018 are two additional ADCP studies on zooplankton in the Mediterranean Sea. Please consider including them.
*A: Done*

R: p.3, l.28: ": : :to infer the composition in the Ligurian Sea: : :". This is incorrect, they only suggest that a change in composition is probable. Please remove.
*A: Done*

R: p.4, l.6-8: Please consider including Brierley et al, Deep-Sea Res., 1998.
*A: Done*

R: p.5, l.18-24: Please consider moving "The operating : : : (Thomson and Emery, 2014)." to introduction and merge the rest of this paragraph with the next one.
*A: Done*

R: p.5, l.32: Do you mean composition instead of ": : :consistency: : :"?
*A: We actually meant consistency (in the sense of texture…).*

R: p.6, l.1-2: "Therefore, : : : quantitative." Repetition (also on p.4, l.6). Please consider removing.
*A: This part has been moved to the introduction, and reworded*

R: p.6, l.5: Please consider moving "The four : : : signals." to the previous paragraph which explain the operation principles.
*A: The previous part has been moved to the introduction, but this is too technical, and has been left in Materials and Methods. However 3.1 and 3.2 has been merged, the new title of this section is "3.1 ADCP settings, and data quality control and estimation of the Mean Volume Backscatter Strength"*

R: p.6, l.6: Please replace ": : :is upward looking: : :" with ": : :is placed at an upward looking position: : :". The way is stated, one might understand that this particular ADCP can be used only in an upward looking position, which is incorrect.
*A: Done*

R: p.6, l.16-20: This paragraph could be removed.
*A: Done*

R: p.6, l.27: Please explain symbols H and θ.
*A: Done*

R: p.7, l.1: There is one PG per transducer and an average PG. Which one was used? The average, the minimum of all separate transducers or something else?
*A: Our data are collected in Earth Coordinates, consequently the four Percent Good values represent (in order):*
*PG1) The percentage of good three-beam solutions (one beam rejected);*
*PG2) The percentage of good transformations (error velocity threshold not exceeded);*
*PG3) The percentage of measurements where more than one beam was bad; and*
*PG4) The percentage of measurements with four-beam solutions.*
*We have used PG4 discarding values below 90%.*

R: p.7, l.13: What data were used for the calculation of the absorption coefficient ?
*A: As it is written in the text, the sound absorption coefficient was computed using a matlab script that needs 3 input parameters: the frequency of the sound pulse in Hz (76800 Hz in this case), temperature in °C (Tx) and pressure (atm). All data were considered at the depth of the ADCP*

R: p.8, l.18: Perhaps you meant ": : :complemented: : :" instead of ": : :integrated: : :"?
*A: Yes you're right*

R: p.8, l.22: Please take also into consideration that large organisms can escape the 200 m mesh. Moriarty et al, Earth Syst. Sci. Data, 2013.
*A: We have mentioned it explicitly in the amended version of the manuscript. at p.8l22 we added: "Some undersampling is possible since large organisms can avoid nets with a small mesh size (Moriarty et al., 2013)."*

R: p.9, l.9: The sentence seems incomplete.
*A: Reviewer 1 suggested modifications, now it should be clearer. Source for moon phases has been added to the list.*

R: p.9, l.12: Please provide more information on spectral analysis. Are the data de-trended, windowed, block-band averaged, which is the number of segments for the spectral estimate, what is the segment overlap? These are necessary for the calculation of confidence intervals.
*A: The spectrum was computed with a straightforward FFT, without segmentation and overlapping. Especially when looking for the long periods, segmentation would not have allowed to detect them. This is why we could not compute the confidence intervals here. We smooth out the text a bit to take into account that we don't know how much we can trust these peaks.*

R: p.9, l.15: Please consider replacing ": : :verify: : :" with ": : :investigate: : :".
*A: Done*

R: p.9, l.18-28: "". Repetition (also on p.4, l.15-24). Please consider removing.
*A: We have cancelled from L21 to 28, the rest was kept to very shortly introduce the Results section*

R: p.10,l.4: Please consider moving the DCM definition to p.5, l.10-15 and add some information regarding its variability from literature.
*A: DCM is a widely accepted definition which refers to the region below the surface of water with the maximum concentration of chlorophyll. We moved its definition to section 3.3. Variability is strongly dependent on season and region, so no general reference concerning it would be meaningful*

R: p.10, l.10: It is not clear to me which this interface is.
*A: It is the interface between AW and IW, in the text we have reworded the sentence to make this clear*

R: p10, l.18-23: I think that "Vertical : : : range." should be moved to methodology.
*A: We left this part here, since it is not really about methods, but what are the implication of this method for the results that we can obtain*

R: p.10, l.26: "All considerations : : : bottom." Repetition (p.7, l.3-4). Please consider removing.
*A: Even in this case we consider that this is important to state, to be clear that we are aware of the limitations that we are faced with.*

R: p.11, l.3: The daily cycle is embedded in the plot, but it is not distinguishable. Please consider including a representative subplot with time span of a few days.
*A: To show this cycle in detail we included the fig.3b to 3g. Another plot would be repetitive*

R: p11, l.7: "June-July 2016". I think it's around April, not June-July.
*A: There was a problem with the labels in Fig. 3b-f, they were misplaced, the correct ones were those of Fig. 3g. So June-July was correct.*

R: p11, l.9: ": : :a pattern : : : organisms." Please add reference.

*A: Done.*

R: p.11, l.22: ": : :daily values are slightly higher than nocturnal values: : :". Please include a supporting graph or mean daily and night MVBS values.
*A: The difference between day and night is well evident from fig. 3b to 3g, because the lines are showing the hour of the day of sunset and sunrise, so "day" is everything between the two black curves, while "night" is everything below the lower curve and above the upper curve. Therefore we think that no supporting additional graph is needed here.*

R: p.11, l.33-34: Please include a plot of integrated MVBS as the argument is not evident from figures 3 - 4.
*A: We think that the Fig. 3a and especially 4a show this, the colourbar shows high values over large parts of the water column between Nov/dec (late fall) and Apr (spring)*

R: p.12, l.3: I think it's ": : :intra-annual: : :" instead of ": : :interannual: : :".
*A: No. we meant interannual, maybe is not ''marked'', we replaced with "clear". The variability is high when you consider the same months of different years and average them, so "interannual" is the right term*

R: p.13, l.5 and figure 4c-4f: Since light intensity was found to be the governing factor controlling DVM (e.g. figure 3), the x axis should be hours relative to sunrise and sunset instead of hour of the day for the W ADCP to be more representative of actual zooplankton migrating velocity. Qualitatively, the results will be the same as those in figure 4d and 4f, but I expect that the duration of upwards and downwards motion will last less time than is shown in the present plot. Please consider, either including a plot with such an x axis, or adding some text explaining that the vertical velocity values are not optimally presented in this plot.
*A: The plot would look like the same, because the x axis represents the actual measurement time, which has a 2-hours interval. Since the plot is an average situation over the whole blooming (or non blooming) we cannot compute the hours relative to sunrise and sunset, because they change every day. An average value of the hour relative to sunrise or sunset would not be so significant and difficult to interpret.*

R: p.14, l.4: ": : :or does not involve a large number of organisms." This applies only (and partially) to MVBS spectra. The way it is stated both MVBS and W are meant, which is incorrect.
*A: We modified the text accordingly*

R: p.14, l.6 and 8: The smallest annotated period in figure 5 is 4.45 hours. It is 4.75 hours in the text.
*A: In the text it is 4.75 hours which is the same as 4 hours and 45 minutes, denoted in the figure.*

R: p.14, l.30-31: "The community is essentially composed by organisms that do not migrate significantly: : :". Please add reference.
*A: We have added "Scotto di Carlo, B., Ianora, A., Fresi, E., and Hure, J.: Vertical zonation patterns for Mediterranean copepods from the surface to 3000 m at a fixed station in the Tyrrhenian Sea, J. Plankton Res., 6, 1031–1056, 1984."*

R: p.15, l.10: Please replace ": : :which is accompanied: : :" with ": : :which is possibly accompanied: : :" as the lack of surface data hampers further investigation.
*A: Done*

R: p.15, l.11-12: Please add reference.
*A: Done*

R: p.15, l.17: Is the distinguish between shallow and deep layers based on the photic layer depth or on another criterion?
*A: We used a simple depth criterion, as it is written in the text, with the data we have we could not use any other definition. The water column was split in half, and the limit of 200m corresponds to about the interface*

*between AW and IW (the values 73m, 201, 378m are the mean bin depths that were used to divide between surface and deep layers).*

R: p.15, l.27: ": : :that well correlate: : :". The Chl- and MVBS time series should be pre-whitened (i.e. remove autocorrelation) before a conclusion is drawn regarding their degree of correlation.
*A: We have now prewhitened the times series (smoothing and detrending), and correlation is even higher, although lags changes a bit. We have modified the text accordingly. Thank you for this comment.*

R: p.15, l.32: Please consider replacing ": : :a surface value: : :" with ": : :an exponentially weighted near-surface value: : :".
*A: Done*

R: p.15-16, l.29-13: Please consider placing the part of literature that supports/contradicts the findings of this study at the beginning of the paragraph and then present possible explanations for this agreement/disagreement. I was confused.
*A: We have rearranged this part*

R: Conclusions section: I think that this part should be rewritten to avoid repetition of results. Instead, the relation of the results of the present study with the relevant literature should be stressed. Also, the length of this section should be substantially shortened.
*A: We have shortened and rewritten some parts of the conclusions*

R: figure 1: Please include information about the data set of SST field in the data availability section.
*A: Done*

R: figure 4a and 4b: The x axis is month or climatological month? It was not clear to me from the text. If climatological please add this to the axis label. Otherwise, state which year the plot refers to.
*A: They are the monthly averages of all available data (average of all Januaries, of all Februaries…). However this is not a proper climatological value. In the caption it is written "monthly mean". We have added this also in the text in this version.*

R: figure 5: Please add confidence intervals. This is particularly important for the low-passed series (5c and 5d) and subsequent interpretation of results.
*A: The spectrum was computed with a straightforward FFT, without segmentation and overlapping. Especially when looking for the long periods as in Fig. 5c and 5d, segmentation would not have allowed to detect them. This is why we did not compute the confidence intervals here. We smooth out the text a bit to take into account that we don't know how much we can trust these peaks.*

R: figure 6a: It seems redundant to me as the three numbers in this plot are already present in the text.
*A: Ok you're right, we removed it.*

R: figure 7b: Please add confidence intervals.
*A: Done*

R: table 1: Please explain symbols in table caption or replace B, L, D, C with blank distance, etc.
*A: The symbols are already explained in the text were the equations are presented. No need for repetition is therefore required according to us.*

R: table 2: I think that this table is redundant, as the useful information of taxonomic analysis has been already presented in the text. Please consider removing.
*A: We decided to keep this table, to summarize the net sample findings, which are not entirely described in the text.*

**Technical corrections**

| Comments | Correction |
|---|---|
| Please add the data availability section that is missing (required by journal). | *Done* |
| Bibliography is not formatted according to journal standards. Number of volume and pages are missing. Also, doi representation is not consistent among references (some are doi:: : : others are https://doi...). | *Done* |
| p.3, l.16: Please merge the two sentences or rephrase. | *Rephrased. Done.* |
| p.4, l.5: ": : :by the depth of the depth: : :". Typo. | *Done.* |
| p.4, l.7: ": : :attempts to calibration: : :". Typo. | *Done.* |
| p.4, l.17: ": : :Data collections: : :". Typo. | *"collected".* |
| p.4, l.22: Perhaps replace ": : :is completed by: : :" with ": : :concludes with: : :"? | *Modified* |
| p.5, l.18: Please replace ": : :as sediments: : :" with ": : :such as sediments: : :". Typo. | *Done.* |
| p.6, l.5: ": : :increments to each other: : :". Typo. | *Done.* |
| p.6, l.28: #7 deployment is missing. Typo. | *Done.* |
| p.7, l.3: Perhaps replace ": : :will be done: : :" with ": : :will be made: : :"? | *Done.* |
| p.9, l.15: Please consider replacing ": : :results to be: : :" with ": : :is a relevant: : :". | *I replaced.* |
| p.9, l.31: Please replace ": : :servicing: : :" with ": : :mooring maintenance: : :". | *Done.* |
| p.9-10, l.31-3: please consider merging the two sentences. | *We prefer to keep the sentences separated* |
| p.10, l.1: ": : :are representative of: : :" instead of ": : :represents: : :". Typo. | *Done.* |
| p.10, l.4: ": : :in correspondence with: : :" instead of ": : :in correspondence of: : :". Typo. | *Done.* |
| p.11, l.24: Perhaps replace ": : :much lower." with ": : :much weaker."? | *As suggested by both reviewers we replaced with "hardly seen"* |
| p.14, l.4: ": : :take place: : :" instead of ": : :take places: : :". Typo. | *Done.* |
| p.14, l.24: Please consider replacing ": : :by far the most abundant group were the copepods: : :" with ": : :the copepods were by far the most abundant: : :". | *Ok* |
| p.14, l.29: Please consider removing "more" in ": : :more western: : :". | *Done.* |
| p.15, l.3: Please consider replacing ": : :more superficial: : :" with "shallower". | *Done.* |
| p.15, l.9: "Fig." instead of "fig.". Typo. | *Done.* |
| p.15, l.26: ": : :are shown: : :" instead of ": : :is shown: : :". Typo. | *Done.* |
| figure 1: Please change "IW=Intermediate Water" to "IW=Intermediate Water path-way" or something similar. | *Done* |
| table 1. ": : :-400 mis: : :". Typo. | *Done. "400 m is".* |

---

## Author Response (AR2)

Revised Submission

| Comments | Corrections |
|---|---|
| Page 1 Line 11. should read "zooplankton" instead of "zooplanktonic organisms" | Done |
| P1L20 remove extra parenthesis "( , …" | Done |
| In abstract you refer to "primary production estimated with satellite data", and continue stating at P1L20 "…with primary production peaks preceding…". But in reality you do not provide any primary production estimates. Most likely you mean " …phytoplankton biomass peaks preceding…". Correct these sentences accordingly, remove references to primary production, unless you can show the data, and refer to phytoplankton biomass. Applies also to P9 L19 | We have replaced this with "biomass of primary producers". At page 9 we wrote "if in this area the phytoplankton biomass is a relevant driver for blooms of secondary producers" |
| P2L10-11: Fractal sentence, please rewrite e.g. "DVM is found within practically all taxonomic zooplankton groups and it is generally assumed that there must be a common reason for such behaviour" | Changed in 'DVM is widespread and found within practically all taxonomic zooplankton groups, so that it is generally assumed that there must be a common underlying reason for such behaviour'' |
| P3L7: Provide a scientific reference instead of the one from instrument manufacturer | It is quite common in papers using ADCP data to mention this reference of the manufacturer technical manual, and we already did it in many other works, since these aspects are really dealt in detail in there. |
| P5 L17: I'm not a specialist regarding the MED water masses, but it sounds strange to state that IW (found at 150-450 m depths) is the "warmest" water mass of MED. | we remove warmest (it is "relatively" warm, i.e. in the TS diagram it is characterized by a relative T max, but depending on the season the AW might obviously become warmer) |
| P6 L7: provide information on ADCP manufacturer | RDI is the name of the manufacturer, we added the whole name in parentheses |
| P6 L19: Should read "ADCP settings" | Done |
| P6 L 24: Give a unit for R (m?) | Done |
| P6 L27: Give a unit for B (m?) | Done |
| P7 L1: Give a unit for H (m?) | Done |
| P8 L6: Should read "…Wetlabs fluorescence sensor …" | Done |
| P13 L30 Onwards and Figure 5: Isn't it a bit strange that differences between observed peak frequencies are exactly – strictly exactly – the same (1.157 * 10^-5 Hz). Please check if there is an artefact in your analyses which creates such harmony. | Our time series consist of 2-hourly data, each data point comes out of an average over 2 hours of 10 seconds measurements. If some organisms move at 4am, another group at 4.45am and a third group at 5am, their signal in W and MVBS are all going into the same average, in the same point of the time series, so |

| | |
|---|---|
| | this explains why they might all end up in the same peak |
| P15 L30: It is clear that primary production – C-fixation by phytoplankton – is the reason for phytoplankton blooms. Therefore, though you have not estimated primary nor secondary production as such, much of the reasoning in caption 4.4. is valid. In P15L30, however, you make very specific statement "the peaks in primary production precede the peaks in secondary production by about three and a half weeks", which is not backed up by data, you have not measured production but biomass. Biomass is not equal to production. Thus modify accordingly. | We replaced "production" with "biomass", thanks! |
| P15 L32: As above | Modified accordingly |
| P16 L2: As above | Modified accordingly |
| P16 L21: As above | Modified accordingly |
| Figure 2. Be consistent in the labels for chlorophyll fluorescence between figures 2d and 2g and figure caption. Fluorescence is a proxy for the concentration, thus 2g is not showing "concentrations". | Done |
| Figure 3. The black lines for sunset and sunrise are hardly visible, could you increase the linewidth | Done |
| Figure 7. Explain briefly in figure caption how Chla estimate was obtained. | Done |
| Please do not refer to primary production when you actually should refer to phytoplankotn biomass or Chlorophyll | Done, see previous answers |
| Check your FFT analyses. It seems very strange if there are multiple biological phenomena (several species in concert!) showing together such an exact mathematical rhythm. | Done, see previous answer |